# Large Language Model Compression with Global Rank and Sparsity Optimization

**Changhai Zhou**[1,5]   **Qian Qiao**[3,5]   **Yuhua Zhou**[4,5]   **Yuxin Wu**[6]   **Shichao Weng**[1,5]
**Weizhong Zhang**[2,*]   **Cheng Jin**[1]
[1]College of Computer Science and Artificial Intelligence, Fudan University
[2]School of Data Science, Fudan University
[3]Soul AILab   [4]Zhejiang University   [5]OpenWPLab   [6]Renmin University of China
chzhou25@m.fudan.edu.cn   {weizhongzhang,jc}@fudan.edu.cn

## Abstract

Low-rank and sparse composite approximation is a natural idea to compress Large Language Models (LLMs). However, such an idea faces two primary challenges that adversely affect the performance of existing methods. The first challenge relates to the interaction and cooperation between low-rank and sparse matrices, while the second involves determining weight allocation across different layers, as redundancy varies considerably among them. To address these challenges, we propose a novel two-stage LLM compression method with the capability of global resource allocation for rank and sparsity. It is noteworthy that the optimization space is vast, making comprehensive optimization computationally prohibitive. Therefore, to reduce the optimization space, our first stage utilizes robust principal component analysis to decompose the weight matrices of LLMs into low-rank and sparse components, which span the low dimensional and sparse spaces containing the resultant low-rank and sparse matrices, respectively. In the second stage, we propose a probabilistic global allocation strategy to jointly identify the low-rank and sparse structures within the above two spaces. The appealing feature of our approach is its ability to automatically detect the redundancy across different layers and to manage the interaction between the sparse and low-rank components. Extensive experimental results indicate that our method significantly surpasses state-of-the-art techniques for sparsification and composite approximation.

## 1 Introduction

Transformer-based large language models (LLMs) (Vaswani et al., 2023; Touvron et al., 2023b; OpenAI et al., 2024) have achieved remarkable progress across natural language processing (NLP), computer vision, and scientific applications. Despite these successes, their massive parameter sizes pose critical challenges: they demand huge storage and memory footprints, incur slow inference, and require substantial computational resources for training. Consequently, model compression (Cheng et al., 2020; Wang et al., 2024a; Zhu et al., 2024; Zhou et al., 2025b; 2026) has become an essential line of research for enabling real-world LLM deployment under stringent hardware constraints.

Among compression strategies, *quantization* (Han et al., 2015; Chee et al., 2023; Kuzmin et al., 2023) typically retains overall model structure by reducing the precision of weights, thus often preserving performance. By contrast, *pruning* (Liu et al., 2017; Frankle & Carbin, 2019; Sun et al., 2024; Frantar & Alistarh, 2023) removes individual weights based on certain criteria (e.g., magnitude or importance scores). Although pruning is flexible and can yield substantial parameter savings, it may degrade performance unless combined with additional fine-tuning or distillation (Sanh et al., 2020; Zhou et al., 2024; 2025a), especially in large-scale LLMs that encode extensive linguistic and factual knowledge (Geva et al., 2021; Dai et al., 2022; Cui et al., 2025).

To retain more critical information under aggressive compression, researchers have explored "low-rank plus sparse" decompositions (Li et al., 2023; Ren & Zhu, 2023; Han et al., 2024). In this

---

*Corresponding author.

approach, the weight matrix is decomposed into a low-rank part that captures global correlations and a sparse part that highlights outliers or domain-specific knowledge. However, existing methods often rely on manually set singular-value thresholds, which can inadvertently discard medium-sized yet important singular values. Additionally, these methods typically require computationally expensive backpropagation for parameter updates. While there is some interaction between the optimization of the low-rank and sparse components, the two parts are still relatively independent in their update processes. Lastly, due to the significant redundancy variations from early layers to deeper ones, how to allocate rank and sparsity across layers in a globally coordinated manner remains unclear.

In this paper, we address these issues via a novel **two-stage compression** framework tailored to LLMs. First, we apply robust principal component analysis (RPCA) (Candès et al., 2011) to factor each weight matrix into strictly low-rank and sparse components, thereby reducing the otherwise huge search space into a low-dimensional subspace and a sparse subspace. Second, we introduce a probabilistic global resource allocation scheme that jointly determines which singular values in the low-rank component and which nonzero entries in the sparse component should be retained. This is done by assigning Bernoulli probabilities and updating them via policy gradient (Williams, 1992) on a small calibration set, avoiding heuristic thresholds or backpropagation on the original LLM parameters. Critically, our method automatically detects the differing redundancy levels across layers and manages the interaction between low-rank and sparse parts, ensuring that vital parameters are kept while redundant ones are pruned away. We summarize our main contributions as follows:

- We propose a two-stage LLM compression approach that first uses RPCA to produce low-rank and sparse subspaces, then employs a Bernoulli-based global resource allocation for rank and sparsity selection.
- Our framework eliminates the need for manual thresholds or layerwise iterative backpropagation, offering a training-free scheme that adapts automatically to various layers' redundancy characteristics.
- Extensive experiments show that our method outperforms existing sparsification and composite approximation baselines under multiple compression ratios, highlighting its effectiveness and robustness.

We provide a detailed review and discussion of related work in Appendix B.

## 2 METHOD

### 2.1 THEORETICAL BACKGROUND AND MOTIVATION

Low-rank approximation is a fundamental technique in matrix theory, widely used to reduce the parameter count in neural networks while preserving model performance. In LLMs, weight matrices are typically high-dimensional and dense. By approximating a weight matrix $W \in \mathbb{R}^{m \times n}$ with rank $R \ll \min(m, n)$ using a truncated SVD, one can write

$$W \approx U_R \Sigma_R V_R^\top, \tag{1}$$

where $U_R$ and $V_R$ contain the top $R$ left and right singular vectors, and $\Sigma_R$ is the diagonal matrix of the largest $R$ singular values. This factorization reduces the parameter count from $m \times n$ to $(m + n) \times R$, and breaks a large matrix multiplication into smaller ones, leading to significant efficiency gains.

Despite these benefits, low-rank approximation alone may be insufficient for LLM compression, especially when the singular values do not decay sharply. For example, Figure 3 in Appendix C shows the singular value spectra of two representative layers (Layer 0 and Layer 31) from a Transformer model, comparing the original weight matrix and its low-rank component after RPCA processing. The dashed lines (original matrices) indicate that certain modules in the same Transformer block (e.g., an attention head vs. a feed-forward network) can exhibit similar spectral shapes; yet across different layers, the redundancy patterns vary considerably. Consequently, imposing the same target rank $R$ across all layers may prune too aggressively in some cases and insufficiently in others. This observation motivates a more flexible approach that can adapt the compression ratio per layer.

Recent studies have explored combining low-rank and sparse representations to enhance compression. For instance, LoSparse (Li et al., 2023) first applies SVD on $W$ to obtain a rank-$R$ approx-

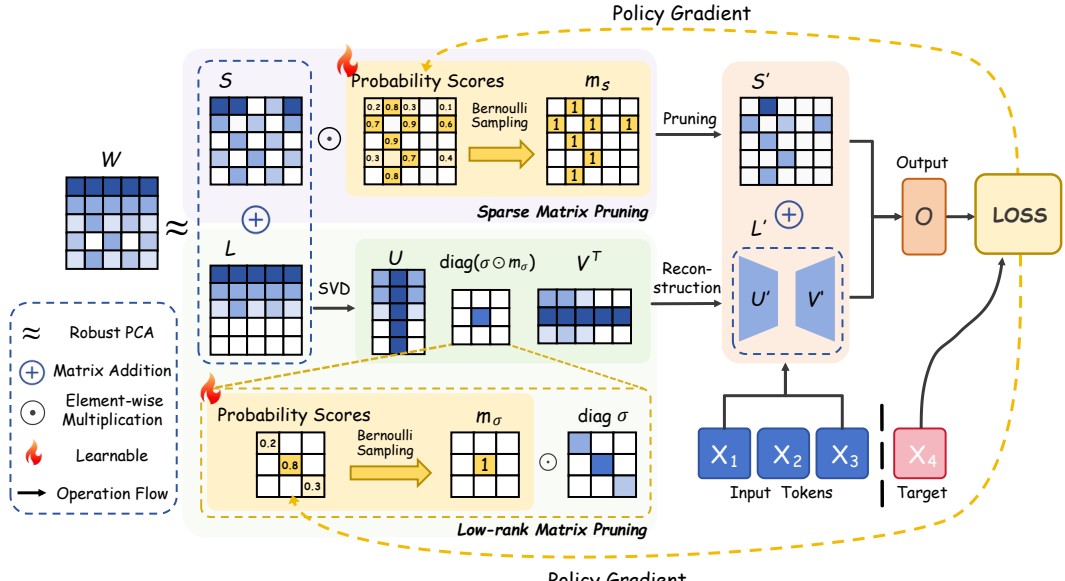

Figure 1: Overview of our proposed compression method. The weight matrix $\mathbf{W}$ is decomposed into a low-rank component $\mathbf{L}$ and a sparse component $\mathbf{S}$ using RPCA. Both components are pruned through Bernoulli sampling guided by learned probability scores, optimized via policy gradient. The low-rank component is further factorized into $\mathbf{U}'$ and $\mathbf{V}'$ to reduce the number of model parameters.

imation, then prunes the residual $W - U_R \Sigma_R V_R^\top$ to form a sparse matrix. In practice, one must still decide the singular value cutoff (or target rank) and the sparsity ratio for the residual. Often, additional fine-tuning is performed on the low-rank part to recover lost performance, or iterative pruning is applied to the sparse part (Molchanov et al., 2019), which can be computationally expensive. A major limitation of these approaches is the reliance on manually chosen thresholds for both singular values and residual pruning. They also lack a clear mechanism to coordinate how much rank vs. sparsity each layer should receive, since different layers and modules may have different redundancy characteristics. Furthermore, when both the low-rank and sparse matrices require joint fine-tuning, the memory consumption can become large, potentially exceeding the budget.

We begin by formulating the global resource allocation objective of compressing LLM weights under a parameter budget (§ 2.2). We then describe our proposed approach (§ 2.3), which first uses RPCA to decompose each weight matrix into low-rank and sparse components, and subsequently prunes these components in a probabilistic manner, without heuristic thresholds or backpropagation through the original LLM weights. Additional theoretical analysis can be found in the Appendix E.

## 2.2 PROBLEM FORMULATION

Suppose we have $L$ layers in an LLM, each containing weight matrices $\{\mathbf{W}^{(l)}\}_{l=1}^L$. We seek compressed matrices $\{\tilde{\mathbf{W}}^{(l)}\}$ such that the total parameter count does not exceed a budget $K$, while minimizing a loss $\ell(\tilde{\mathbf{W}})$ measured on a small calibration set $\mathcal{D}$. Formally,

$$\min_{\{\tilde{\mathbf{W}}^{(l)}\}} \quad \sum_{(x,y)\in\mathcal{D}} \ell\Big(f(\tilde{\mathbf{W}};x),\, y\Big),$$

$$\text{subject to} \quad \text{ParamCount}\big(\{\tilde{\mathbf{W}}^{(l)}\}\big) \;\leq\; K, \tag{2}$$

where $f(\tilde{\mathbf{W}};x)$ is the LLM's forward pass given the compressed weights, and $\text{ParamCount}(\cdot)$ measures how many parameters are retained. Directly pruning each individual weight is intractable for very large matrices. To address this, we propose to:

- **Decompose** each $\mathbf{W}^{(l)}$ via RPCA to obtain a low-rank matrix $\mathbf{L}$ and a sparse matrix $\mathbf{S}$, reducing the search space to "global rank directions" plus "sparse outliers."

- **Probabilistically prune** both components under the budget $K$ by learning Bernoulli retention probabilities through policy gradient on a small calibration set.

## 2.3 PROPOSED APPROACH: CAP

As illustrated in Figure 1, our proposed method, CAP, follows a two-stage process. The role of Stage 1 is to decompose weights into a relatively low-rank matrix $\mathbf{L}$ and a sparse matrix $\mathbf{S}$, reducing the parameter space to manageable candidates. Stage 2 then allocates the parameter budget over these candidates to achieve the target compression ratio while preserving model performance. This principled decomposition followed by budget-aware selection avoids heuristic thresholds and expensive fine-tuning. In the following sections, we provide detailed explanations of our algorithm.

### 2.3.1 STAGE 1: PRINCIPLED DECOMPOSITION VIA RPCA

The first stage of our method is not designed to achieve a target compression ratio directly. Instead, its purpose is to perform a principled decomposition of each weight matrix, transforming the complex problem of pruning individual weights into a more structured one. By separating a weight matrix $\mathbf{W} \in \mathbb{R}^{m \times n}$ into a low-rank component $\mathbf{L}$ that captures global structure and a sparse component $\mathbf{S}$ that captures local, salient features, we establish a high-quality candidate pool for subsequent compression. We achieve this through RPCA, which formulates the decomposition as a convex optimization problem:

$$\min_{\mathbf{L},\mathbf{S}} \quad \underbrace{\|\mathbf{L}\|_*}_{\text{Low-rank constraint}} \quad + \lambda \quad \underbrace{\|\mathbf{S}\|_1}_{\text{Sparsity constraint}} \quad \text{subject to} \, \mathbf{W} = \mathbf{L} + \mathbf{S}. \tag{3}$$

The choice of this objective is theoretically motivated. The nuclear norm $\|\mathbf{L}\|_*$ is the tightest convex relaxation of the rank function, making it the most effective convex proxy for minimizing rank. Similarly, the $\ell_1$ norm $\|\mathbf{S}\|_1$ is the standard convex relaxation for the non-convex $\ell_0$ norm (sparsity), which effectively identifies significant, sparse outliers. Thus, this framework provides a principled and globally optimal separation of $\mathbf{W}$ into its underlying low-rank and sparse structures.

Crucially, the hyperparameter $\lambda$ in the RPCA objective governs the nature of this decomposition, not the final compression rate. Attempting to control sparsity by simply tuning $\lambda$ leads to unpredictable changes in the rank of $\mathbf{L}$ and often results in poor-quality decompositions, a point we analyze in detail in Appendix H. Therefore, this stage focuses solely on creating an optimal candidate pool for the subsequent budget-aware pruning. We solve Eq. equation 3 using the efficient Alternating Direction Method of Multipliers (ADMM) (Lin et al., 2010). The updates are as follows:

$$\mathbf{L}_{k+1} = \arg\min_{\mathbf{L}} \, \|\mathbf{L}\|_* + \frac{\mu}{2} \left\| \mathbf{W} - \mathbf{L} - \mathbf{S}_k + \mu^{-1}\mathbf{Y}_k \right\|_F^2, \tag{4}$$

$$\mathbf{S}_{k+1} = \arg\min_{\mathbf{S}} \, \lambda\|\mathbf{S}\|_1 + \frac{\mu}{2} \left\| \mathbf{W} - \mathbf{L}_{k+1} - \mathbf{S} + \mu^{-1}\mathbf{Y}_k \right\|_F^2, \tag{5}$$

$$\mathbf{Y}_{k+1} = \mathbf{Y}_k + \mu \left( \mathbf{W} - \mathbf{L}_{k+1} - \mathbf{S}_{k+1} \right). \tag{6}$$

The $\mathbf{L}$-update employs Singular Value Thresholding (SVT) (Cai et al., 2008):

$$\mathbf{L}_{k+1} = \mathbf{U} \, \text{diag}(\text{shrink}_{\mu^{-1}}(\boldsymbol{\sigma})) \mathbf{V}^\top \tag{7}$$

where $\mathbf{U}\boldsymbol{\sigma}\mathbf{V}^\top$ is the SVD of $\mathbf{W} - \mathbf{S}_k + \mu^{-1}\mathbf{Y}_k$, with singular value shrinkage $\text{shrink}_\tau(\sigma_i) = \max(\sigma_i - \tau, 0)$. The $\mathbf{S}$-update applies elementwise soft-thresholding:

$$[\mathbf{S}_{k+1}]_{ij} = \text{shrink}_{\lambda\mu^{-1}}([\mathbf{W} - \mathbf{L}_{k+1} + \mu^{-1}\mathbf{Y}_k]_{ij}) \tag{8}$$

This alternating optimization progressively separates the weight matrix into a low-dimensional subspace capturing directional patterns ($\mathbf{L}$) and a sparse subspace containing localized refinements ($\mathbf{S}$), establishing the foundation for subsequent global resource allocation.

### 2.3.2 STAGE 2: LEARNABLE PROBABILISTIC PRUNING

While the RPCA decomposition in Stage 1 provides a high-quality separation of components, it does not enforce a specific parameter budget. The second stage directly addresses this by performing a global, budget-aware selection from the candidate pools ($\mathbf{L}$ and $\mathbf{S}$) generated previously. We decide

which rank-1 components in $\mathbf{L}$ and which non-zero entries in $\mathbf{S}$ to keep, to meet a user-defined parameter budget $K$ while minimizing task performance degradation.

The total parameter budget, $K$, is a user-defined hyperparameter (e.g., 50% of the original model's parameters). Each retained singular value $\sigma_i$ from $\mathbf{L}$ requires storing its corresponding singular vectors $\mathbf{u}_i \in \mathbb{R}^m$ and $\mathbf{v}_i \in \mathbb{R}^n$, contributing $(m+n)$ parameters. Each retained non-zero entry of $\mathbf{S}$ contributes one parameter. We introduce Bernoulli random variables to model the retention decision for each potential parameter:

$$m_{\sigma_i} \sim \text{Bernoulli}(s_{\sigma_i}), \quad m_{S_{ij}} \sim \text{Bernoulli}(s_{S_{ij}}),$$

where $s_{\sigma_i} \in [0,1]$ and $s_{S_{ij}} \in [0,1]$ are learned retention probabilities. The compressed matrix is then

$$\tilde{\mathbf{W}} = \mathbf{U}\,\text{diag}(\boldsymbol{\sigma} \odot \mathbf{m}_\sigma)\,\mathbf{V}^\top + \mathbf{S} \odot \mathbf{m}_S, \tag{9}$$

subject to $\sum_i s_{\sigma_i}(m+n) + \sum_{i,j} s_{S_{ij}} \leq K$ to respect the total parameter budget.

**Learning probabilities via policy gradient.** We minimize the expected loss on a small calibration set $\mathcal{D}$:

$$\min_{\mathbf{s}}\ \mathbb{E}_{\mathbf{m}\sim p(\mathbf{m}|\mathbf{s})}\Big[\mathcal{L}(\tilde{\mathbf{W}})\Big], \tag{10}$$

where $\mathbf{s} = \{s_{\sigma_i}, s_{S_{ij}}\}$ and $p(\mathbf{m} \mid \mathbf{s})$ is the product of Bernoulli distributions. We employ a REINFORCE-style (Williams, 1992) policy gradient:

$$\nabla_{s_k} \mathbb{E}_{\mathbf{m}}[\mathcal{L}(\tilde{\mathbf{W}})] = \mathbb{E}_{\mathbf{m}}\Big[\mathcal{L}(\tilde{\mathbf{W}})\,\nabla_{s_k}\log p(\mathbf{m} \mid s_k)\Big]. \tag{11}$$

For a Bernoulli variable $m_k \sim \text{Bernoulli}(s_k)$,

$$\nabla_{s_k}\log p(m_k \mid s_k) = \frac{m_k - s_k}{s_k(1-s_k) + \epsilon},$$

with a small $\epsilon > 0$ to avoid division by zero. To reduce variance, we maintain a moving average baseline $\delta$ (Zhao et al., 2011):

$$\delta \leftarrow \beta\,\delta + (1-\beta)\,\mathcal{L}(\tilde{\mathbf{W}}), \tag{12}$$

and update each $s_k$ via

$$s_k \leftarrow s_k - \eta\big(\mathcal{L}(\tilde{\mathbf{W}}) - \delta\big)\nabla_{s_k}\log p(m_k \mid s_k). \tag{13}$$

After each gradient step, we project $\mathbf{s}$ back onto $\{\mathbf{s} : \mathbf{1}^\top\mathbf{s} \leq K,\ 0 \leq s_k \leq 1\}$.

**Thresholding masks and final factorization.** The policy gradient optimization yields a set of probabilities $\{s_k\}$ that reflect the learned importance of each parameter for minimizing the task loss. To obtain the final compressed model that strictly adheres to the budget $K$, we perform a deterministic selection. We treat the probabilities $s_k$ as importance scores for their corresponding parameters (singular values or sparse entries). All potential parameters are ranked globally according to these scores. We select the top-$K$ parameters to keep, generating the final binary masks $m_k$:

$$m_k = \begin{cases} 1, & \text{if parameter } k \text{ is among the top-}K \text{ scored parameters}, \\ 0, & \text{otherwise.} \end{cases} \tag{14}$$

We note that comparing $s_k$ across different parameter types is valid because the learned probability acts as a unified proxy for the "utility-to-cost" ratio. During optimization, the policy gradient inherently accounts for the parameter's contribution to loss reduction relative to its existence in the model. Thus, sorting by $s_k$ provides a globally consistent ranking for resource allocation.

This final step ensures the parameter budget is met precisely. The compressed weight matrix is reconstructed using these binary masks in Eq. equation 9. To enhance efficiency, the resulting low-rank component is factorized into smaller matrices. The compressed $\mathbf{U}'$ and $\mathbf{V}'$ are computed as:

$$\mathbf{U}' = [\sqrt{\sigma_1}\mathbf{u}_1,\ \sqrt{\sigma_2}\mathbf{u}_2,\ \ldots,\ \sqrt{\sigma_{r'}}\mathbf{u}_{r'}], \tag{15}$$

$$\mathbf{V}' = [\sqrt{\sigma_1}\mathbf{v}_1,\ \sqrt{\sigma_2}\mathbf{v}_2,\ \ldots,\ \sqrt{\sigma_{r'}}\mathbf{v}_{r'}], \tag{16}$$

where $r'$ is the number of retained singular values (i.e., where $m_{\sigma_i} = 1$). The final compressed weight matrix is then:

$$\tilde{\mathbf{W}} = \mathbf{U}'(\mathbf{V}')^\top + \mathbf{S} \odot \mathbf{m}_S. \tag{17}$$

This factorization reduces both storage and computational cost during inference.

Table 1: Performance comparison with unstructured pruning methods at 50% compression. We report average zero-shot accuracy (%) across eight tasks and WikiText perplexity (lower is better).

| Method | Compression | Zero-shot Accuracy (%) | | | | WikiText Perplexity | | | |
|---|---|---|---|---|---|---|---|---|---|
| | | Phi-3 Mini | Phi-3 Medium | LLaMA-3 8B | LLaMA-3 70B | Phi-3 Mini | Phi-3 Medium | LLaMA-3 8B | LLaMA-3 70B |
| Dense | 0% | 72.85 | 75.37 | 70.79 | 76.53 | 9.42 | 6.11 | 8.56 | 2.68 |
| *Uniform Sparsity Methods* | | | | | | | | | |
| SparseGPT | 30% | 70.63 | 74.53 | 69.08 | 75.07 | 11.19 | 7.48 | 9.71 | 3.24 |
| | 40% | 69.18 | 74.40 | 67.58 | 74.63 | 13.03 | 8.52 | 10.01 | 3.99 |
| | 50% | 66.36 | 73.25 | 64.66 | 73.17 | 16.80 | 9.89 | 11.95 | 5.27 |
| Wanda | 30% | 70.66 | 74.05 | 68.63 | 75.19 | 10.71 | 7.28 | 9.39 | 3.28 |
| | 40% | 68.80 | 73.01 | 67.04 | 74.10 | 12.59 | 8.49 | 9.74 | 4.08 |
| | 50% | 65.03 | 70.96 | 63.27 | 72.85 | 17.23 | 10.12 | 12.36 | 5.38 |
| DSNoT | 30% | 71.20 | 74.03 | 68.98 | 75.54 | 10.51 | 7.11 | 9.36 | 3.27 |
| | 40% | 69.08 | 72.90 | 66.65 | 74.29 | 12.17 | 8.24 | 9.60 | 4.10 |
| | 50% | 65.33 | 71.12 | 62.74 | 72.91 | 16.68 | 9.96 | 12.41 | 5.58 |
| OATS | 30% | 71.48 | 74.04 | 69.34 | 75.24 | 10.27 | 6.85 | 9.59 | 3.07 |
| | 40% | 70.04 | 74.46 | 68.68 | 74.88 | 11.53 | 7.70 | 9.24 | 3.68 |
| | 50% | 68.41 | 73.39 | 65.71 | 73.30 | 15.18 | 9.05 | 10.87 | 4.78 |
| *Layerwise Allocation Methods (Based on Wanda)* | | | | | | | | | |
| OWL | 30% | 71.15 | 74.28 | 69.12 | 75.45 | 10.45 | 7.15 | 9.25 | 3.18 |
| | 40% | 69.32 | 73.35 | 67.58 | 74.42 | 12.28 | 8.32 | 9.58 | 3.95 |
| | 50% | 65.78 | 71.38 | 63.95 | 73.25 | 16.85 | 9.88 | 12.18 | 5.25 |
| AlphaPruning | 30% | 71.28 | 74.35 | 69.25 | 75.52 | 10.38 | 7.08 | 9.18 | 3.15 |
| | 40% | 69.45 | 73.48 | 67.72 | 74.55 | 12.15 | 8.25 | 9.48 | 3.88 |
| | 50% | 65.95 | 71.52 | 64.12 | 73.42 | 16.72 | 9.78 | 12.05 | 5.18 |
| *Our Method* | | | | | | | | | |
| **CAP** | 30% | **72.15** | **74.85** | **70.25** | **76.02** | **9.88** | **6.58** | **9.05** | **2.95** |
| | 40% | **70.58** | **74.78** | **69.38** | **75.45** | **11.15** | **7.42** | **9.16** | **3.52** |
| | 50% | **69.12** | **74.05** | **66.85** | **74.18** | **14.68** | **8.78** | **10.35** | **4.45** |

## 2.4 DISCUSSION

We propose CAP, a two-stage compression framework for large language models. Stage 1: RPCA Decomposition—The weight matrix is split into low-rank and sparse parts, preserving global structure while isolating local anomalies and sharply reducing the search space for later optimization. This step is cast a convex program (nuclear norm + $L_1$ norm), guaranteeing a globally optimal separation for the decomposition objective. Stage 2: Bernoulli Mask Optimization—Using a small calibration set, an unbiased policy-gradient method learns the retention probabilities for the low-rank and sparse components, automatically detecting and pruning redundancy across layers.

**Remark.** While Stage 1 solves a convex problem, Stage 2 addresses the discrete budget allocation problem via policy gradient, which serves as a heuristic optimizer. It does not guarantee a global optimum for the non-convex pruning objective but empirically finds effective allocation strategies by leveraging the high-quality subspaces identified in Stage 1. This two-step design balances theoretical principledness in decomposition with practical flexibility in resource allocation.

## 3 EXPERIMENTS

In this section, we first introduce the experimental setup. Subsequently, we present the main experimental results, extensions to modern instruction-tuned models, and detailed ablation studies. We further provide in-depth analyses on inference throughput and calibration robustness. Due to space constraints, detailed results for Llama-1/2 are provided in Appendix I, and thorough analyses of throughput performance and computational resource consumption of our proposed **CAP** method are presented in Appendix J.

**Models and Evaluation.** We evaluate our proposed CAP method on a comprehensive set of widely adopted large language models across different architectures and scales. Our evaluation includes the LLaMA family: LLaMA-1 (Touvron et al., 2023a) (7B, 13B, 30B), LLaMA-2 (Touvron et al., 2023b) (7B, 13B), and LLaMA-3 (Dubey et al., 2024) (8B, 70B); Modern Instruction-Tuned models: LLaMA-3.1-8B and Qwen2.5-7B (Yang et al., 2024); the OPT series (Zhang et al., 2022) (1.3B, 2.7B, 6.7B, 13B); the Phi-3 family (Abdin et al., 2024) including Phi-3 Mini (3.8B) and Phi-3 Medium (14B); and BERT-base (Devlin et al., 2019). To assess the performance of the compressed models, we conduct experiments on zero-shot tasks and language modeling. We perform an extensive evaluation of the zero-shot capabilities of pruned models across eight standard commonsense benchmark datasets: GLUE (Wang et al., 2019a), PIQA (Bisk et al., 2020), BoolQ (Clark et al., 2019), HellaSwag (Zellers et al., 2019), WinoGrande (Sakaguchi et al., 2021), OpenBookQA (Mi-

Table 2: Performance on Modern LLMs (50% Sparsity). Left: LLaMA-3.1-8B-Instruct on reasoning and long-context tasks. Right: Qwen2.5-7B on diverse benchmarks.

| | LLaMA-3.1-8B-Instruct | | | | Qwen2.5-7B | | | |
|---|---|---|---|---|---|---|---|---|
| **Method** | **GSM8K** (8-shot, %) | **LongBench-v2** (Avg, %) | **WikiText** (PPL) | **Task** | **SparseGPT** (50%) | **Wanda** (50%) | **CAP** (50%) | **Dense** (0%) |
| Dense | 84.5 | 30.4 | 6.01 | NarrativeQA | 4.95 | 16.30 | **18.52** | 31.9 |
| Wanda (50%) | 45.6 | 25.1 | 7.26 | GovReport | 33.52 | 32.13 | **34.05** | 53.4 |
| **CAP (50%)** | **56.8** | **27.2** | **6.61** | Lcc (Code) | 39.70 | 45.14 | **46.92** | 58.1 |
| *Improvement* | *+11.2* | *+2.1* | *-0.65* | TriviaQA | 89.35 | 88.70 | **89.85** | 92.5 |

haylov et al., 2018), and the ARC Easy and ARC Challenge tasks (Clark et al., 2018). For language modeling evaluation, we measure perplexity on the held-out WikiText (Merity et al., 2016) validation set (lm-eval-harness). For modern models, we additionally evaluate Chain-of-Thought reasoning (GSM8K (Bai et al., 2025)) and LongBench-v2 (Cobbe et al., 2021).

**Implementation Details.** We utilize PyTorch 2.3.0, Transformers 4.28.0, CUDA 12.1 on NVIDIA A100 GPUs under Ubuntu. To ensure fair comparison, we use 128 sequences with context length sampled from the C4 training set (Raffel et al., 2020) as calibration data. For policy gradient estimation, we set iterations to 3, sliding window size to 5, and learning rate to 0.05. The $\lambda$ parameter for RPCA decomposition is set according to the established formulation $\lambda = 1/\sqrt{\max(m,n)}$, where $m$ and $n$ represent the dimensions of the data matrix. Further ablation studies on the $\lambda$ setting can be found in Appendix H.

**Baselines.** We compare our approach with several compression techniques: **SparseGPT** (Frantar & Alistarh, 2023) is a second-order pruning method for LLMs that solves a layer-wise reconstruction problem. **WANDA** (Sun et al., 2024) prunes weights based on their estimated importance using activation statistics. **OATS** (Zhang & Papyan, 2024) performs optimal sparsity allocation across transformer layers using second-order information. **OWL** (Yin et al., 2023) and **AlphaPruning** (Lu et al., 2024) are layer-wise allocation methods that optimize sparsity distribution. **SLiM** (Mozaffari et al., 2024) combines low-rank approximation with sparsity and quantization, featuring probabilistic quantization error fitting. **LPAF** (Ren & Zhu, 2023) first applies first-order unstructured pruning to obtain a low-rank sparse model. Then, sparsity-aware SVD is used to decompose the sparse matrices into a low-rank form **AB**. Finally, mixed-rank fine-tuning is used to retrain **AB**. We also compare against dense SVD-based methods **SVD-LLM2** (Wang et al., 2025c), **Dobi-SVD** (Wang et al., 2025b), **Basis Sharing** (Wang et al., 2025a) and structured pruning method **LoSparse** (Li et al., 2023).

## 3.1 COMPARISON WITH UNSTRUCTURED PRUNING METHODS

We compare CAP with recent unstructured pruning methods across multiple large language models. Table 1 presents a comprehensive comparison including both uniform sparsity methods (SparseGPT, Wanda, DSNoT, OATS) and layerwise allocation methods (OWL, AlphaPruning) at 30%, 40%, and 50% compression ratios. Note that OWL and AlphaPruning are layerwise allocation methods that optimize sparsity distribution across layers, and we implement them using Wanda as the base pruning method for fair comparison. CAP consistently achieves competitive or superior performance across different model architectures and sizes.

## 3.2 EXTENSION TO MODERN INSTRUCTION-TUNED MODELS

To validate CAP's effectiveness on the latest generation of models, we evaluated **LLaMA-3.1-8B-Instruct** and **Qwen2.5-7B** at 50% sparsity. We report on Chain-of-Thought reasoning (GSM8K) and long-context understanding (LongBench-v2), where standard pruning methods struggle. As shown in Table 2, CAP significantly outperforms Wanda on challenging tasks. For LLaMA-3.1-Instruct, CAP recovers **+11.2%** accuracy on GSM8K, suggesting that the preserved low-rank backbone is crucial for maintaining the precise reasoning circuits disrupted by unstructured pruning.

Table 3: Comparison at 50% unstructured sparsity. Zero-shot accuracy (%) on representative models. LoRA variants: Naive-LoRA uses basic error compensation; SLiM-LoRA incorporates weight salience; SLiM-LoRAQ additionally quantizes the adapter.

| Method | Quantization | OPT | | | | LLaMA-2 | |
|---|---|---|---|---|---|---|---|
| | | 1.3B | 2.7B | 6.7B | 13B | 7B | 13B |
| Dense | - | 43.4 | 45.5 | 48.3 | 48.7 | 56.6 | 60.8 |
| Magnitude | Group AbsMax | 32.1 | 39.9 | 36.4 | 32.3 | 47.0 | 51.0 |
| SparseGPT | OPTQ | 38.7 | 43.4 | 47.0 | 47.4 | 51.1 | 55.9 |
| Wanda | OPTQ | 41.0 | 42.9 | 46.5 | 46.8 | 53.6 | 56.8 |
| JSQ | JSQ | 38.9 | 35.5 | 42.8 | 30.7 | 52.3 | 57.0 |
| L2QER | Group AbsMax | 38.4 | 41.3 | 45.1 | OOM | 50.6 | OOM |
| Naive-LoRA | QuantizationW | 40.4 | 43.4 | 46.6 | 47.3 | 51.5 | 55.3 |
| SLiM-LoRA | QuantizationW | **41.9** | 43.5 | 47.1 | 48.0 | 54.3 | 57.9 |
| SLiM-LoRAQ | QuantizationW | 41.7 | 43.6 | 47.2 | 47.9 | 54.2 | 57.3 |
| **CAP (Ours)** | OPTQ | 41.7 | **44.8** | **48.2** | **48.3** | **55.1** | **59.2** |

## 3.3 COMPARISON WITH JOINT COMPRESSION METHODS

Since methods like LoSparse are based on structured pruning and require extensive retraining, we compare CAP with SLiM, a state-of-the-art method that jointly applies quantization, sparsity, and low-rank approximation. We also include comparisons with other joint compression approaches including JSQ (Guo et al., 2024), a joint sparsity and quantization method that optimizes sparsity and quantization parameters simultaneously, and L2QER (Zhang et al., 2024a), which combines low-rank decomposition, quantization, and sparsity in a sequential manner.

While both SLiM and CAP structurally combine low-rank and sparse components, their technical approaches differ fundamentally: SLiM primarily focuses on using low-rank decomposition to fit quantization errors through probabilistic reformulation and numerical integration to find optimal quantization parameters, whereas CAP focuses on the synergy between low-rank and sparse decomposition through RPCA, where the low-rank component emerges from joint optimization rather than serving as an error fitting tool. Table 3 presents the comparison on representative models at 50% unstructured sparsity. The results demonstrate that CAP consistently outperforms existing joint compression methods across different model sizes and architectures.

## 3.4 COMPREHENSIVE COMPARISON ON GLUE TASKS

We evaluate CAP on downstream tasks using the GLUE benchmark with BERT-base. Table 4 compares CAP against various compression paradigms including pre-training distillation, task-specific distillation, structured pruning, and matrix factorization methods.

CAP achieves competitive or superior performance across most GLUE tasks and compression ratios. Notably, CAP consistently outperforms methods without fine-tuning and achieves comparable results to fine-tuned methods like LPAF while using only the RPCA decomposition without additional task-specific fine-tuning.

## 3.5 COMPARISON WITH STRUCTURED AND PURE LOW-RANK METHODS

We additionally compare CAP against (1) pure low-rank decomposition methods, including **SVD-LLM v2** (Wang et al., 2025c), **Dobi-SVD** (Wang et al., 2025b), and **Basis Sharing** (Wang et al., 2025a), which utilize dense or shared SVD structures; and (2) **LoSparse** (Li et al., 2023), a representative structured method requiring iterative fine-tuning.

As shown in Table 5 (Left), CAP yields significantly better perplexity than pure SVD methods (e.g., 5.85 vs 6.08 at 20% ratio), confirming the necessity of the sparse component. Table 5 (Right) shows that while our training-free CAP is competitive, applying fine-tuning (CAP w/ FT) significantly outperforms LoSparse, validating RPCA as a superior initialization.

Table 4: Results on GLUE tasks under different parameter budgets. We show accuracy (%) for RTE, MRPC, SST-2, QNLI, MNLI and F1 score (%) for QQP.

| Method | RTE 50% | 25% | 16% | MRPC 50% | 25% | 16% | SST-2 50% | 25% | 16% | QQP 50% | 25% | 16% | QNLI 50% | 25% | 16% | MNLI 50% | 25% | 16% |
|---|---|---|---|---|---|---|---|---|---|---|---|---|---|---|---|---|---|---|
| **Pre-training Distillation** | | | | | | | | | | | | | | | | | | |
| DistilBERT | 65.0 | 61.0 | 56.3 | 85.8 | 77.0 | 72.5 | 90.0 | 88.9 | 86.4 | 90.8 | 89.4 | 88.0 | 86.0 | 83.8 | 81.6 | 81.7 | 76.4 | 71.3 |
| TinyBERT | 67.7 | 67.2 | 64.6 | 86.3 | 85.3 | 78.2 | 92.3 | 89.8 | 88.0 | 90.5 | 90.0 | 88.7 | 89.9 | 87.7 | 84.5 | 83.1 | 80.6 | 77.4 |
| **Task-specific Distillation** | | | | | | | | | | | | | | | | | | |
| PKD | 65.5 | 59.2 | 53.8 | 81.9 | 76.2 | 71.3 | 91.3 | 88.1 | 87.2 | 88.4 | 88.5 | 87.5 | 88.4 | 82.7 | 78.0 | 81.3 | 75.7 | 72.7 |
| Theseus | 65.6 | 62.1 | 58.8 | 86.2 | 77.2 | 72.8 | 91.5 | 88.6 | 86.1 | 90.9 | 89.6 | 89.0 | 88.2 | 83.2 | 78.0 | 82.3 | 76.4 | 73.5 |
| CKD | 67.3 | 66.5 | 60.8 | 86.0 | 81.1 | 76.6 | 91.2 | 90.0 | 88.7 | 90.5 | 88.7 | 89.5 | 90.4 | 86.4 | 81.9 | 83.5 | 79.0 | 76.8 |
| MetaDistill | 69.0 | 66.7 | 61.0 | 86.8 | 81.8 | 77.3 | 92.3 | 88.9 | 87.0 | 91.0 | 88.9 | 86.9 | 90.4 | 86.8 | 84.9 | 83.5 | 79.5 | 76.8 |
| **Structured Pruning** | | | | | | | | | | | | | | | | | | |
| ISP | 66.4 | 65.0 | 63.9 | 86.1 | 83.6 | 82.8 | 90.4 | 89.4 | 89.9 | 90.5 | 88.7 | 87.2 | 90.5 | 88.7 | 87.2 | 83.2 | 81.9 | 80.8 |
| FLOP | 66.1 | 58.5 | 56.0 | 82.1 | 80.1 | 78.4 | 89.7 | 89.1 | 87.9 | 91.4 | 89.9 | 89.7 | 90.5 | 88.5 | 87.1 | 82.6 | 79.9 | 79.0 |
| BPhybrid | 66.4 | 64.3 | 63.9 | 84.1 | 81.1 | 78.3 | 91.0 | 88.7 | 86.9 | 91.8 | 89.3 | 89.1 | 90.7 | 88.1 | 86.2 | 83.0 | 80.1 | 78.0 |
| CoFi | 69.0 | 66.4 | 66.4 | 84.6 | 84.3 | 83.4 | 91.6 | 89.7 | 89.2 | 90.1 | 89.0 | 88.9 | 90.2 | 88.8 | 87.6 | 83.5 | 80.8 | 80.5 |
| **Matrix Factorization** | | | | | | | | | | | | | | | | | | |
| SVD$_{ft}$ | 62.1 | 60.3 | 55.6 | 79.9 | 77.0 | 70.1 | 89.4 | 86.9 | 85.3 | 90.0 | 87.9 | 87.1 | 90.1 | 83.8 | 80.9 | 81.8 | 78.0 | 74.6 |
| LPAF | 62.8 | 68.0 | 67.9 | 86.8 | 85.5 | 86.0 | 92.0 | 90.0 | 91.5 | 90.4 | 90.1 | 91.1 | 89.3 | 88.6 | 84.8 | 84.8 | 82.6 | 77.6 |
| **Low-rank plus Sparse** | | | | | | | | | | | | | | | | | | |
| **CAP (Ours)** | **69.1** | 67.8 | 66.5 | 86.2 | **86.2** | 85.8 | **92.3** | **91.9** | 90.8 | **91.9** | 90.8 | 90.5 | **90.8** | 89.1 | 88.8 | **85.1** | 83.1 | 82.8 |
| **BERT-base** | | 69.2 | | | 86.4 | | | 92.7 | | | 91.5 | | | 91.4 | | | 84.6 | |

Table 5: Left: Comparison with Post-Training SVD Methods on LLaMA-7B (WikiText-2 PPL). Right: Comparison with LoSparse on DeBERTa-V3-base (20% Retention).

(a) SVD Methods Comparison (PPL)

| Method | Format | 20% | 40% | 60% |
|---|---|---|---|---|
| SVD-LLM v2 | Dense LR | 7.12 | 10.34 | 14.71 |
| Dobi-SVD | Dense LR | 6.08 | 8.48 | 15.62 |
| Basis Sharing | Shared LR | 7.74 | 12.39 | 28.72 |
| **CAP (Ours)** | **LR+Sparse** | **5.85** | **6.39** | **7.06** |

(b) LoSparse Comparison (Accuracy)

| Method | FT? | MNLI | QNLI |
|---|---|---|---|
| LoSparse | Yes | 83.8 | 88.6 |
| CAP (Ours) | No | 78.2 | 83.0 |
| **CAP (Ours)** | **Yes** | **86.8** | **91.8** |

## 3.6 IN-DEPTH ANALYSIS: EFFICIENCY AND ROBUSTNESS

Table 6: Left: Inference Efficiency on LLaMA-3.1-8B (A100-80G). Right: Calibration Set Robustness (LLaMA-3.1-8B, 50% Sparsity).

| Method | S-Sparsity | Latency | Throughput |
|---|---|---|---|
| Wanda (50%) | 50% | 6.28 ms | 163.4 tok/s |
| **CAP (50%)** | **∼85%** | **5.80 ms** | **176.5 tok/s** |

| Calib. Set | Domain | PPL | Acc. |
|---|---|---|---|
| **C4 (Default)** | General | **7.05** | **76.8** |
| WikiText-2 | Formal | 6.88 | 75.4 |
| GitHub Code | Code | 7.18 | 76.1 |

We investigate the practical inference efficiency and the robustness of our calibration process. Table 6 (Left) shows that CAP achieves higher throughput (176.5 tok/s) than Wanda (163.4 tok/s). This is because CAP's sparse component $S$ falls into the high-sparsity regime ($> 85\%$), where SpMM is highly efficient compared to the uniform 50% sparsity of standard pruning methods. Table 6 (Right) analyzes the impact of different calibration datasets. While using in-domain formal text (WikiText-2) yields the lowest perplexity (6.88), the diverse C4 dataset provides the best generalization capability with the highest zero-shot accuracy (76.8%). Notably, the model demonstrates strong robustness: even when calibrated on a distinct domain like GitHub Code, the accuracy remains competitive (76.1%).

## 3.7 ABLATION STUDIES

To gain deeper insights into the behavior of our compression method, we conduct ablation studies focusing on two key aspects: (i) the distribution of different matrix ranks after compression is

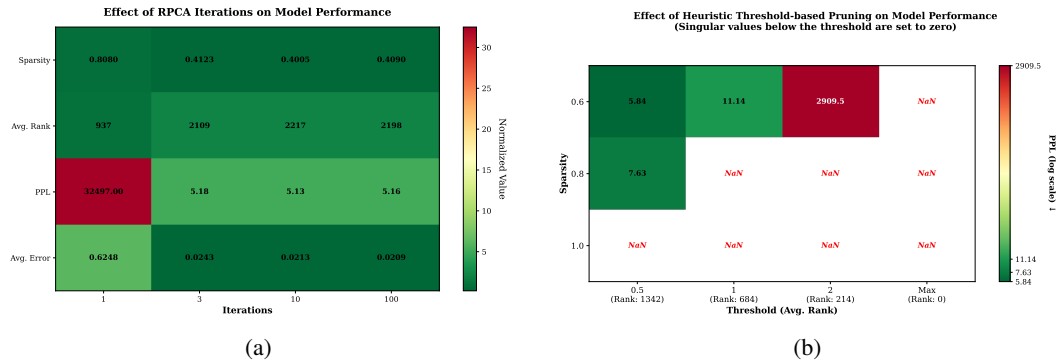

Figure 2: (a) Effect of RPCA iterations on model performance. Here, "Avg. Rank" denotes the average rank of the low-rank component, and "Sparsity" represents the sparsity of the sparse component. (b) Effect of heuristic threshold-based pruning on model performance. Singular values below the threshold are set to zero.

between 200 and 800; and (ii) the stability of our method when pruning is applied sequentially layer-by-layer. Detailed analysis and experimental results are provided in Appendix K.

**Robustness and Rapid Convergence of RPCA Decomposition** We investigated the effect of RPCA iterations on the performance of the LLaMA2-7B model to assess the robustness of the decomposition quality. Figure 2a shows that only a few RPCA iterations are needed to achieve an effective decomposition, providing a solid and stable starting point for subsequent pruning. **This rapid convergence demonstrates the robustness of the RPCA stage**, as it consistently produces a high-quality separation of global patterns (low-rank component) and local anomalies (sparse component) across different layers with minimal computational overhead. Additionally, the "Avg. Error" column represents the average approximation error for each matrix, offering insight into the model's tolerance to error. Similar to findings in the quantization field, large models exhibit robustness to approximation errors. The fact that performance even surpasses the original model after decomposition further underscores the effectiveness of RPCA in identifying and isolating redundant parameters, thereby enhancing the input quality for the subsequent global optimization stage.

**Necessity of Global Resource Allocation** The limitations of heuristic, post-decomposition pruning underscore the importance of our proposed global optimization components (policy gradient with Bernoulli sampling). We conducted experiments using a uniform threshold-based approach applied to the RPCA output. In this method, we prune the low-rank component $\mathbf{L}$ by setting singular values below a specific threshold to zero and remove low-magnitude elements from the sparse component $\mathbf{S}$ without applying our probabilistic masking or additional optimization. Figure 2b summarizes the results for LLaMA2-7B, which indicate that both components are indispensable: retaining only one leads to a performance collapse. **This clear failure of simple thresholding strategies validates our core design choice: the necessity of a learned, global resource allocation strategy**. Unlike rigid heuristics, policy gradient optimization and Bernoulli sampling mechanism determine the rank and sparsity allocation across layers based on their redundancy characteristics, which is crucial for maintaining model performance under compression.

## 4 CONCLUSION

In this work, we introduce CAP, a novel two-stage compression framework for large language models. First, CAP applies Robust Principal Component Analysis (RPCA) to decompose weight matrices into low-rank and sparse subspaces, drastically reducing the optimization space. Second, it employs a **global resource allocation strategy** via policy gradients to jointly optimize parameter retention under a strict budget. Unlike heuristic-based approaches, CAP adaptively allocates rank and sparsity **in a training-free manner**, avoiding expensive backpropagation on the original weights. Extensive evaluations on models like LLaMA-3.1 and Qwen2.5 demonstrate that CAP significantly outperforms SOTA baselines in reasoning and long-context tasks.

ACKNOWLEDGEMENT

This work was supported by the National Nature Science Foundation of China (62472097), Shanghai Municipal Science and Technology Commission (Grant No.25511102200), and Fudan Kunpeng&Ascend Center of Cultivation. The computations in this research were performed on the CFFF platform of Fudan University.

REPRODUCIBILITY STATEMENT

This statement presents a comprehensive report detailing the reproduction process for our RPCA-based model compression methodology, incorporating policy gradient optimization. The implementation builds upon WANDA's code base and integrates components from additional open-source libraries, to which we extend our gratitude.

IMPLEMENTATION OVERVIEW

The proposed algorithm is implemented using PyTorch and Hugging Face's Transformers library. The core components of the implementation include:

- **RPCA Decomposition**: Each weight matrix $\mathbf{W}$ from the pre-trained model is decomposed into a low-rank matrix $\mathbf{L}$ and a sparse matrix $\mathbf{S}$ using Robust Principal Component Analysis (RPCA). This decomposition captures global structure in $\mathbf{L}$ and local anomalies in $\mathbf{S}$.
- **Probabilistic Pruning**: Bernoulli random variables are introduced to determine the retention of singular values in $\mathbf{L}$ and specific elements in $\mathbf{S}$. Retention probabilities are treated as trainable parameters.
- **Policy Gradient Optimization**: A policy gradient framework optimizes the retention probabilities by minimizing the expected loss over a calibration dataset, subject to a parameter budget constraint.
- **Model Reconstruction**: Following optimization, compressed weight matrices are reconstructed using the retained components. Low-rank matrices are further factorized to enhance computational efficiency during inference.

CODE STRUCTURE

The implementation is organized into three main components:

- `main.py`: The primary entry point for the pruning process, handling model loading, argument parsing, and execution.
- `lib/prune_rl.py`: Contains the RPCA decomposition, policy gradient optimization routines, and model reconstruction logic.
- `main.sh`: A shell script to streamline the pruning execution process with preset arguments.

RUNNING THE PRUNING PROCESS

To reproduce our results, follow these steps:

1. **Environment Setup**:
   - Ensure Python 3.8 or later is installed.
   - Install the necessary dependencies:
     ```
     pip install torch \
       transformers \
       numpy \
       tqdm \
       matplotlib \
     ```

```
json \
argparse
```

2. **Execution**:

   - Use the provided shell script `main.sh` to execute the pruning process with preset configurations:

     ```
     bash main.sh
     ```

   - The script handles model selection, pruning method, RPCA parameters, policy gradient settings, and output configurations.

## KEY IMPLEMENTATION DETAILS

- **Code Base**: The implementation builds upon WANDA's pruning framework, modified to incorporate RPCA decomposition and policy gradient optimization.
- **RPCA Implementation**: An augmented Lagrange multiplier method is used to solve the RPCA optimization problem. This separates the weight matrix into $\mathbf{L}$ and $\mathbf{S}$, capturing essential patterns and anomalies, respectively.
- **Bernoulli Masks**: For each singular value in $\mathbf{L}$ and each element in $\mathbf{S}$, a Bernoulli random variable determines its retention. Retention probabilities are initialized uniformly and optimized iteratively.
- **Policy Gradient Optimization**: Retention probabilities are refined using a policy gradient approach. The gradients of the expected loss with respect to the probabilities are estimated and used to update the masks, with variance reduced via a moving average baseline.
- **Model Reconstruction**: Following optimization, probabilities are thresholded to generate binary masks. The compressed model is reconstructed, and low-rank matrices are further decomposed into $\mathbf{U}'$ and $\mathbf{V}'$ for inference efficiency.

## RESULTS

Using the aforementioned process, we successfully compressed the LLaMA-2-7B model to achieve a 50% compression rate while maintaining performance. Perplexity was monitored after processing each layer to evaluate the model's performance.

## CONCLUSION

This reproduction report outlines the implementation and procedural details for replicating our RPCA-based compression method with policy gradient optimization. The provided code base, built upon WANDA, ensures reproducibility and offers a robust foundation for advancing model compression research.

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

# Table of Contents

## A  THE USE OF LARGE LANGUAGE MODELS(LLMS)

In preparing this paper, LLMs were employed solely for language refinementpurposes, such as improving grammar, clarity, and style of expression. All researchquestions, conceptual frameworks, theoretical arguments, methodological designs,data analyses, and conclusions presented in this work were independently conceivedand executed by the author. The LLMs did not generate, alter, or influence theunderlying ideas, interpretations, or findings. Their use was limited to assistingin polishing the readability and fluency of the manuscript while preserving theoriginality and integrity of the scholarly contributions.

## B  RELATED WORK

### B.1  UNSTRUCTURED PRUNING

Unstructured pruning eliminates individual weights by setting them to zero, providing fine-grained control over model sparsity. **SparseGPT** (Frantar & Alistarh, 2023) leverages second-order information to perform layer-wise pruning with minimal retraining, whereas **Wanda** (Sun et al., 2024) combines weight magnitude with activation statistics for a more straightforward pruning strategy. While these methods achieve efficient pruning, they struggle to maintain performance at high sparsity levels and often require additional retraining (Sanh et al., 2020; Renda et al., 2020). **BESA** (Xu et al., 2024) introduces a differentiable pruning framework that dynamically allocates sparsity across layers to minimize performance degradation, producing competitive results without requiring extensive retraining. **Dynamic Sparse Training (DST)** (Liu et al., 2020) proposes an end-to-end sparse training method where trainable pruning thresholds dynamically adjust the sparsity level during training. Unlike post-training pruning methods, DST eliminates the need for iterative fine-tuning by continuously optimizing layer-wise sparsity using backpropagation. DST is designed for training sparse networks from scratch.

More recent methods focus on improving sparsity allocation across layers. **OATS** (Zhang & Papyan, 2024) formulates optimal sparsity allocation as a constrained optimization problem using second-order sensitivity (Hessian-based), enabling non-uniform sparsity distribution across layers while preserving overall model accuracy. Similarly, **DSNoT** (Zhang et al., 2024b) proposes a data-free unstructured pruning method that identifies salient weights using gradient sign stability, making it suitable for low-resource settings. To further enhance performance, several approaches explore adaptive layer-wise sparsity. **OWL** (Yin et al., 2023) and **AlphaPruning** (Lu et al., 2024) both leverage activation statistics—such as outlier magnitudes or sparsity patterns—to determine how much sparsity each layer can tolerate, thereby optimizing the global sparsity budget. These methods typically build upon simpler base pruners like Wanda and improve performance by reallocating sparsity in a layer-dependent manner. In contrast, our method **CAP** performs joint low-rank and sparse decomposition via RPCA, which naturally induces structured sparsity patterns and enables global optimization through policy gradients, avoiding hand-crafted allocation heuristics.

Low-rank approximation, obtained via truncated SVD, remains a cornerstone for reducing both memory footprint and FLOPs in deep networks (Denton et al., 2014). Early composite schemes such as **LoSparse** (Li et al., 2023) add a sparse "correction" to each low-rank factor, but depend on hand-tuned singular-value cut-offs and iterative fine-tuning. **LPAF** (Ren & Zhu, 2023) improves robustness by applying structured pruning first, then decomposing the residual with a sparsity-aware SVD and mixed-rank re-training ($W \approx AB$), yet still requires several post-processing passes. Recent work pushes the idea further: **SVD-LLM** (Wang et al., 2024b) introduces a truncation-aware criterion that keeps LLaMA-13B perplexity at 6.43 with only 20% of the weights, while **MoDeGPT** (Lin et al., 2025) performs modular low-rank decomposition across consecutive sub-layers and preserves 90–95% zero-shot accuracy at 25–30% parameters *without any fine-tuning*. On the sparse side, **MaskLLM** (Fang et al., 2024) learns hardware-friendly 2:4 masks, retaining 91–95% of baseline accuracy at 50% sparsity and yielding a $\sim 1.4\times$ speed-up on A100 GPUs.

Hybrid methods combine sparsity with quantization or low precision: **SpQR** (Dettmers et al., 2024) stores a tiny FP16 outlier matrix plus 4-bit weights, achieving sub-1% perplexity loss, while **CALDERA** (Saha et al., 2024) represents each layer as a low-rank term plus a quantized backbone, pushing below 3 bits/parameter on models up to 70B.

Recent advances integrate low-rank adaptation with sparsity and quantization. For instance, **SLiM** (Mozaffari et al., 2024) combines low-rank modules, unstructured sparsity, and quantization, introducing a probabilistic framework to fit quantization errors using low-rank components. It further proposes SLiM-LoRA variants that apply sparsity-aware adapters with salience-based compensation. Similarly, **JSQ** (Guo et al., 2024) jointly optimizes sparsity and quantization parameters through a unified objective, while **L2QER** (Zhang et al., 2024a) adopts a sequential pipeline of low-rank decomposition, sparsification, and quantization to maximize compression efficiency. These methods demonstrate the growing trend toward multi-modal compression. However, most rely on heuristic designs or require multiple stages of fine-tuning.

Unlike the above approaches, our **CAP** framework uses *Robust PCA* to *jointly* discover layer-wise low-rank and sparse subspaces, then optimizes Bernoulli masks globally via policy gradients—eliminating manual thresholds and any backpropagation over the original parameters. This enables a training-free, end-to-end decomposition that unifies the benefits of low-rank structure and sparse expressivity without relying on error-fitting or staged optimization.

### B.2 MODEL COMPRESSION VIA DISTILLATION AND STRUCTURED PRUNING

Knowledge distillation transfers knowledge from a large teacher model to a smaller student through output mimicking or intermediate feature alignment. Early works such as **DistilBERT** (Sanh et al., 2019) and **TinyBERT** (Jiao et al., 2019) apply distillation during pre-training, while task-specific variants like **PKD** (Sun et al., 2019) and **Theseus** (Xu et al., 2020) focus on fine-tuned compression. More advanced frameworks such as **CKD** (Mirzadeh et al., 2020) and **MetaDistill** (Zhou et al., 2022) introduce multi-stage or meta-learning strategies to improve distillation efficiency. On the pruning side, structured methods remove entire neurons, heads, or blocks. **ISP** (McCarley, 2019) and **FLOP** (Prasanna et al., 2020) use importance scoring for layer pruning, while **BPhybrid** (Lagunas et al., 2021) combines block pruning with fine-tuning. **CoFi** (Xia et al., 2022) jointly prunes weights and attention heads using a shared importance metric. Unlike these methods that require extensive fine-tuning or teacher models, our approach operates in a post-training, training-free manner, making it more suitable for low-resource deployment scenarios.

## C PRELIMINARIES

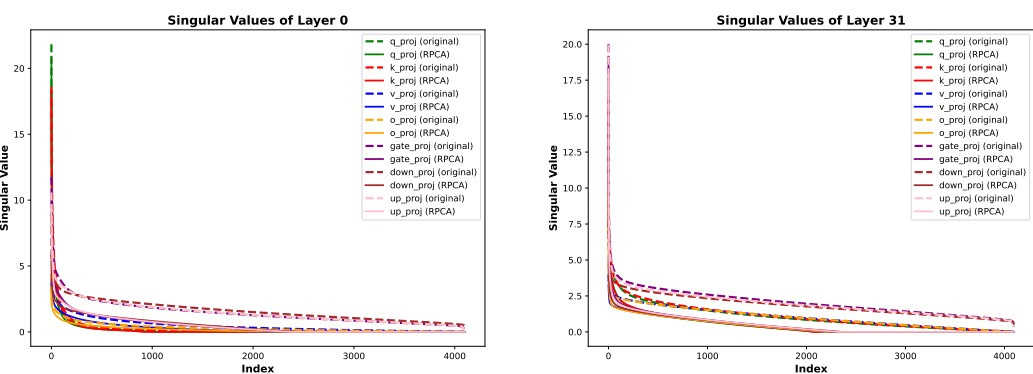

Figure 3: Singular values of Layer 0 and Layer 31 across different modules, comparing original and RPCA-processed matrices. The dotted line represents the singular value distribution of the original model, and the solid line represents the singular value distribution of the low-rank matrix after RPCA processing.

### C.1 LOW-RANK APPROXIMATION

Low-rank approximation (Chu et al., 2003) is a fundamental technique in matrix theory, widely used to reduce the parameter count in neural networks while preserving most of the model's performance. In large language models (LLMs), weight matrices are typically high-dimensional and dense. By

approximating a weight matrix $\mathbf{W} \in \mathbb{R}^{M \times N}$ as $\mathbf{U}\mathbf{V}^\top$ with rank $R \ll \min(M, N)$, we can achieve substantial reductions in storage and computational cost. Concretely, one typically uses *Singular Value Decomposition* (SVD) to write

$$\mathbf{W} = \mathbf{U}\boldsymbol{\Sigma}\mathbf{V}^\top, \tag{18}$$

and then retains only the largest $R$ singular values $\boldsymbol{\Sigma}_R$, yielding

$$\mathbf{W} \approx \mathbf{U}_R \, \boldsymbol{\Sigma}_R \, \mathbf{V}_R^\top. \tag{19}$$

This factorization can reduce the parameter count from $M \times N$ to $(M + N) \times R$, and also break a large matrix multiplication into smaller ones:

$$\mathbf{W}\mathbf{x} \approx \mathbf{U}_R\big(\boldsymbol{\Sigma}_R(\mathbf{V}_R^\top \mathbf{x})\big),$$

leading to efficiency gains.

Despite these benefits, low-rank approximation alone may be insufficient for LLM compression, especially when the singular values do not decay sharply. Figures 3 illustrate the singular value distributions for two layers (Layer 0 and Layer 31) in a large Transformer. The dashed lines represent the original matrices, showing that certain modules in the same Transformer block (e.g., attention vs. feedforward) might exhibit similar shapes, yet across different layers, the *redundancy patterns* can vary considerably. Consequently, imposing the same rank $R$ uniformly across all layers may prune too aggressively in some places and insufficiently in others. A more flexible approach is needed to handle these differences among modules and layers.

### C.2 Low-Rank Approximation with Sparse Corrections

To mitigate the shortcomings of purely low-rank approximation, recent methods (Li et al., 2023; Ren & Zhu, 2023) advocate combining a low-rank matrix with a *sparse* correction term. One splits the model weights as:

$$\mathbf{W} = \underbrace{\mathbf{U}\mathbf{V}^\top}_{\text{low-rank}} + \underbrace{\mathbf{S}}_{\text{sparse}}. \tag{20}$$

LoSparse (Li et al., 2023), for example, first applies an SVD on $\mathbf{W}$ to obtain a low-rank component (with some rank $R$), and then prunes the residual $\mathbf{W} - \mathbf{U}\mathbf{V}^\top$ to form a sparse matrix $\mathbf{S}$. In practice, one must still decide the singular-value cutoff (or target rank) and the sparsity ratio for the residual. Often, additional fine-tuning is performed on the low-rank part to recover lost performance, or iterative pruning is applied to the sparse part (Molchanov et al., 2019), which can be computationally expensive.

A major limitation of these approaches is that they rely heavily on *manually specified* thresholds for both singular values and residual pruning. They also lack a clear mechanism to coordinate how much rank vs. how much sparsity each layer should receive, since different layers and modules may have different redundancy patterns. Furthermore, when both the low-rank matrix and the sparse matrix need simultaneous updates (or fine-tuning), memory consumption can become large, often exceeding the budget for fine-tuning.

## D  Knowledge Neurons

Transformer-based architectures, particularly large language models (LLMs), serve as repositories of linguistic and factual knowledge (Geva et al., 2021; Dai et al., 2022). This knowledge is intricately distributed across the network's feed-forward networks (FFNs) and attention mechanisms, forming the basis for accurate language understanding and generation. Figure 4 provides an illustrative depiction of how such knowledge is encoded, stored, and attributed across these components, with factual information such as "Ireland's capital is Dublin" encapsulated through complex interactions.

### D.1  Impact of Pruning FFN Layers

FFNs act as key-value storage within Transformer models, encoding linguistic and factual information as neuron activations (Geva et al., 2021). Specific neurons, often referred to as *knowledge*

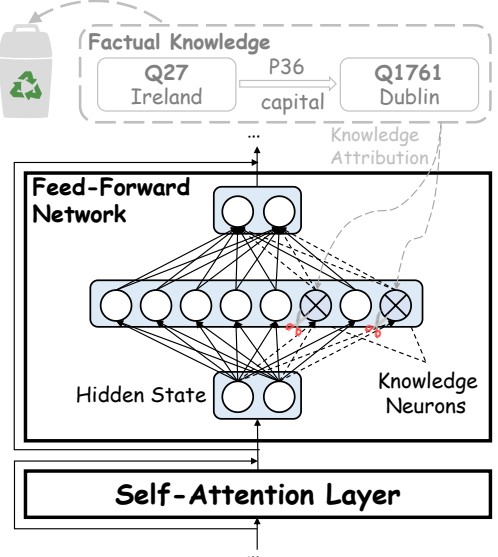

Figure 4: An illustration of how factual knowledge is encoded and attributed within Transformer architectures. Factual knowledge is distributed across feed-forward networks (FFNs) and attention mechanisms. Pruning these components risks disrupting knowledge structures, leading to performance degradation.

*neurons*, are responsible for capturing and representing precise knowledge. For example, one neuron may activate to encode "Q27: Ireland," while its interplay with others encodes the factual relationship "Capital: Dublin."

Pruning FFN layers introduces the following risks:

- **Disruption of Knowledge Neurons:** Pruning weights indiscriminately can remove neurons responsible for encoding critical facts, leading to the loss of semantic consistency and factual integrity.

- **Recovery Complexity:** Unlike structured pruning or quantization, unstructured pruning typically requires extensive fine-tuning to recover lost performance, as critical neurons are often irreversibly removed.

## D.2 EFFECT OF PRUNING ATTENTION MECHANISMS

Attention mechanisms are integral to Transformer models, enabling dynamic token-wise interactions to capture semantic and contextual information. They play two primary roles:

- **Knowledge Attribution:** Attention mechanisms identify and link related entities, such as establishing the factual connection between "Ireland" and "Dublin."

- **Contextual Understanding:** By dynamically weighting token interactions, attention heads provide rich semantic understanding, ensuring the accurate representation of factual relationships.

Pruning attention mechanisms poses distinct challenges:

- **Impaired Attribution:** Removing attention heads or weights can disrupt critical connections between tokens, such as the association between "Ireland" and "Dublin," resulting in factual inconsistencies.

- **Redundancy vs. Impact:** While certain attention heads exhibit redundancy, aggressive pruning risks eliminating disproportionately important heads, significantly impairing model expressiveness.

### D.3 Challenges in Simultaneous Pruning

The concurrent pruning of FFNs and attention mechanisms amplifies risks, as both components play complementary roles. FFNs encode factual knowledge, while attention mechanisms attribute and contextualize this information. Disrupting either component undermines the model's capacity to process and retrieve information effectively. The key challenges include:

- **Degraded Knowledge Retrieval:** Pruning FFNs may impair the model's ability to retrieve stored knowledge, while pruning attention mechanisms compromises its ability to contextualize and attribute this knowledge accurately.
- **Trade-offs in Compression:** Achieving a balance between parameter reduction and knowledge retention demands fine-grained strategies that preserve essential structures while compressing redundant components.

### D.4 Proposed Mitigation Strategies

To address these challenges, we propose a composite approximation framework designed to preserve critical structures within FFNs and attention mechanisms while achieving significant compression:

- **Robust Principal Component Analysis (RPCA):** RPCA decomposes weight matrices into low-rank and sparse components, separating global patterns from local anomalies. This allows us to target redundancies without compromising essential knowledge structures.
- **Policy Gradient Optimization:** By introducing Bernoulli distributions, we selectively retain important components in both FFNs and attention layers. Policy gradient methods efficiently optimize the retention probabilities, bypassing the need for heuristic thresholds.
- **Layer-Adaptive Compression:** Our approach applies module-specific pruning rates, ensuring critical parameters for knowledge retention remain intact while compressing less significant structures.

### D.5 Conclusion

The interplay between FFNs and attention mechanisms highlights their distinct yet complementary roles in encoding and attributing knowledge within Transformer models. While FFNs store knowledge, attention mechanisms enable its contextualization. Effective compression requires strategies that preserve the integrity of these components. Our composite approximation framework achieves this balance by leveraging RPCA and policy-driven optimization, offering a robust solution for retaining critical knowledge while reducing model complexity.

## E  Additional Details and Theoretical Analysis

### E.1  Policy Gradient with Moving Average Baseline

The REINFORCE gradient estimator in Equation equation 11 has high variance because it scales directly with the loss magnitude. We incorporate a moving average baseline $\delta$ to reduce variance while maintaining unbiased estimates:

$$\nabla_{s_k}\mathbb{E}[\mathcal{L}] \approx \mathbb{E}\left[(\mathcal{L}(\tilde{\mathbf{W}}) - \delta)\nabla_{s_k}\log p(m_k|s_k)\right] \tag{21}$$

- **Variance Reduction**: Let $\delta = \mathbb{E}[\mathcal{L}]$ be the expected loss. The variance becomes:

$$\mathrm{Var}[(\mathcal{L} - \delta)\nabla\log p] = \mathrm{Var}[\mathcal{L}\nabla\log p] - 2\delta\mathrm{Cov}(\mathcal{L}\nabla\log p, \nabla\log p)$$
$$+ \delta^2\mathrm{Var}[\nabla\log p]$$

The baseline minimizes the second term when $\delta \approx \mathbb{E}[\mathcal{L}]$ (Zhao et al., 2011).

- **Unbiased Estimation**: The baseline introduces no bias because:

$$\mathbb{E}[\delta\nabla\log p] = \delta\mathbb{E}[\nabla\log p] = 0 \tag{22}$$

- **Practical Implementation**: We update $\delta$ as an exponential moving average (EMA):

$$\delta \leftarrow \beta\delta + (1 - \beta)\mathcal{L}(\tilde{\mathbf{W}}) \tag{23}$$

with $\beta = 0.9$ in experiments. This tracks recent performance while being robust to noise.

## E.2 THEORETICAL ANALYSIS FOR LLM COMPRESSION

**Theorem E.1** (Low-Rank+Sparse Approximation). *For any weight matrix $\mathbf{W} \in \mathbb{R}^{m \times n}$ in Transformer layers, let $r^*$ be the intrinsic rank and $s^*$ the sparsity level. CAP achieves:*

$$\|\tilde{\mathbf{W}} - \mathbf{W}\|_F \leq \underbrace{C\sqrt{\frac{r^*}{m+n}}}_{\text{low-rank error}} + \underbrace{D\sqrt{s^*}}_{\text{sparse error}} + \mathcal{O}\left(\sqrt{\frac{\log(1/\delta)}{|\mathcal{D}|}}\right) \tag{24}$$

*with probability $1 - \delta$, where $C, D$ are data-dependent constants.*

*Proof.* From RPCA recovery bounds (Candès et al., 2011) and PAC-Bayes generalization. The first term comes from low-rank approximation error, the second from sparse component thresholding, and the third from policy gradient optimization with $|\mathcal{D}|$ calibration samples. $\square$

**Lemma E.2** (Parameter Efficiency). *CAP preserves model capacity with:*

$$\text{rank}(\tilde{\mathbf{L}}) = \mathcal{O}\left(\frac{K}{m+n}\right), \quad \|\tilde{\mathbf{S}}\|_0 = \mathcal{O}(K) \tag{25}$$

*where $K$ is the parameter budget. This matches the optimal rates for low-rank + sparse representations.*

**Corollary E.3** (LLM Performance Preservation). *For a Transformer with $L$ layers, if each attention/MLP matrix satisfies Theorem 1 with $\|\tilde{\mathbf{W}}^{(l)} - \mathbf{W}^{(l)}\|_F \leq \epsilon$, then the full model satisfies:*

$$|\mathcal{L}(\tilde{\mathbf{W}}) - \mathcal{L}(\mathbf{W})| \leq L\epsilon\sqrt{\dim(\mathbf{x})} \tag{26}$$

*where $\dim(\mathbf{x})$ is the input dimension.*

# F CONVERGENCE ANALYSIS OF BERNOULLI POLICY GRADIENT

We analyse the stochastic optimisation that drives CAP's second stage and show that it is *(i)* unbiased, *(ii)* has controllable variance, and *(iii)* converges to a local optimum under standard Robbins–Monro conditions.

**Unbiased gradient.** For a scalar loss $L(\widetilde{W})$ and Bernoulli mask vector $m \sim p(m \,|\, s)$, the REINFORCE estimator (Williams, 1992; Baxter & Bartlett, 2001) is

$$g(s) = (L(\widetilde{W}) - \delta)\,\nabla_s \log p(m \,|\, s), \tag{27}$$

giving

$$\mathbb{E}_m\big[g(s)\big] = \nabla_s \mathbb{E}_m[L(\widetilde{W})],$$

so the estimate is unbiased.

**Mask statistics.** Each entry $m_{ij} \sim \text{Bernoulli}(s_{ij})$ satisfies $\mathbb{E}[m_{ij}] = s_{ij}$ and $\text{Var}(m_{ij}) = s_{ij}(1 - s_{ij})$, maximised at $s_{ij} = 0.5$. We mitigate variance via:

- a moving–average baseline $\delta$ in Eq. equation 27, which subtracts an estimate of $\mathbb{E}[L]$;
- mini-batch averaging over $\mathcal{B}$ mask samples, reducing variance by $|\mathcal{B}|^{-1}$.

During training $s_{ij} \to 0$ or 1, so $\text{Var}(m_{ij}) \to 0$ and gradients become increasingly stable.

**Convergence.** With bounded second moment of $g(s)$, step sizes $\eta_t$ satisfying $\sum_t \eta_t = \infty$ and $\sum_t \eta_t^2 < \infty$, the Robbins–Monro theorem ensures almost-sure convergence to a stationary point (Robbins & Monro, 1951). Empirically, CAP converges within $\mathcal{O}(10^3)$ updates on a 128-sample calibration set.

## G    BASELINE METHODS DESCRIPTION

This section provides detailed descriptions of all baseline methods used in our experimental evaluation. We categorize these methods into several groups based on their compression approaches.

### G.1    UNSTRUCTURED PRUNING METHODS

**SparseGPT**    (Frantar & Alistarh, 2023) is a second-order pruning method specifically designed for large language models. It formulates pruning as a layer-wise reconstruction problem, using the inverse Hessian to determine optimal weight removal while minimizing the increase in layer-wise reconstruction error. The method processes weights in each layer sequentially and updates remaining weights to compensate for the removal of pruned parameters.

**WANDA**    (Sun et al., 2024) (Pruning by Weights AND Activations) is a simple yet effective pruning approach that estimates weight importance using both weight magnitudes and activation statistics. It computes importance scores by multiplying weight magnitudes with the norm of corresponding input activations, providing a more comprehensive measure of parameter significance than magnitude-only methods.

**DSNoT**    (Zhang et al., 2024b) (Dual Sparse Network Training) applies structured sparsity patterns during training to achieve efficient inference. The method maintains dual sparse networks during training and applies knowledge distillation between dense and sparse models to preserve performance.

**OATS**    (Zhang & Papyan, 2024) (Optimal Allocation for Transformer Sparsity) performs optimal sparsity allocation across transformer layers using second-order information. It leverages layer-wise sensitivity analysis to determine the optimal distribution of sparsity across different layers, considering the varying importance of different transformer components.

### G.2    LAYER-WISE ALLOCATION METHODS

**OWL**    (Yin et al., 2023) (Outlier-Aware Weight Layerwise) is a layer-wise allocation method that optimizes sparsity distribution across layers by identifying and preserving outlier weights that are critical for model performance. The method uses activation-based outlier detection to guide the sparsity allocation process.

**AlphaPruning**    (Lu et al., 2024) employs reinforcement learning to automatically determine the optimal sparsity ratio for each layer. It formulates the layer-wise sparsity allocation as a sequential decision-making problem and uses policy gradient methods to learn optimal allocation strategies.

### G.3    JOINT COMPRESSION METHODS

**SLiM**    (Mozaffari et al., 2024) (Sparsity-aware Low-rank compression with Importance Masking) combines low-rank approximation with sparsity and quantization. Its key innovation is probabilistic quantization error fitting, where low-rank decomposition is used to model and compensate for quantization errors. The method employs numerical integration to find optimal quantization parameters.

**JSQ**    (Guo et al., 2024) (Joint Sparsity and Quantization) optimizes sparsity and quantization parameters simultaneously through a unified optimization framework. It formulates the compression problem as a joint optimization over both sparsity masks and quantization levels, enabling better trade-offs between compression ratio and model performance.

**L2QER** (Zhang et al., 2024a) (Low-rank Quantization with Error Reduction) combines low-rank decomposition, quantization, and sparsity in a sequential manner. It first applies low-rank decomposition, then quantizes the resulting factors, and finally applies sparsity to further reduce model size while using error compensation techniques.

**LPAF** (Ren & Zhu, 2023) (Low-rank Plus Sparse with Adaptive Fine-tuning) follows a three-stage approach: (1) applies first-order unstructured pruning to obtain a sparse model, (2) uses sparsity-aware Singular Value Decomposition (SVD) to decompose the sparse matrices into low-rank form $\mathbf{AB}$, and (3) performs mixed-rank fine-tuning to retrain the decomposed matrices while preserving the sparse structure.

### G.4 KNOWLEDGE DISTILLATION METHODS

#### G.4.1 PRE-TRAINING DISTILLATION

**DistilBERT** (Sanh et al., 2019) applies knowledge distillation during the pre-training phase to create a smaller model. It uses a combination of distillation loss, masked language modeling loss, and cosine embedding loss to train a student model that retains much of the teacher's capabilities with significantly fewer parameters.

**TinyBERT** (Jiao et al., 2019) extends knowledge distillation by transferring knowledge from both the intermediate layers and the prediction layer of the teacher model. It employs attention-based distillation and hidden state distillation to capture more comprehensive knowledge from the teacher model.

#### G.4.2 TASK-SPECIFIC DISTILLATION

**PKD** (Sun et al., 2019) (Patient Knowledge Distillation) introduces a patient teacher mechanism where the student model learns from multiple intermediate teacher models of varying sizes. This progressive distillation approach helps bridge the capacity gap between large teachers and small students.

**Theseus** (Xu et al., 2020) employs a progressive module replacement strategy during fine-tuning. It gradually replaces modules in the original model with smaller counterparts while maintaining performance through careful scheduling and knowledge transfer.

**CKD** (Mirzadeh et al., 2020) (Cascade Knowledge Distillation) addresses the capacity gap problem in knowledge distillation by introducing intermediate teacher models. It uses a cascade of teacher models with gradually decreasing sizes to provide a smooth knowledge transfer path.

**MetaDistill** (Zhou et al., 2022) leverages meta-learning to automatically discover optimal distillation strategies. It learns to adapt distillation parameters and strategies based on the specific characteristics of the teacher-student pair and the target task.

### G.5 STRUCTURED PRUNING METHODS

**ISP** (McCarley, 2019) (Iterative Structured Pruning) applies structured pruning in an iterative manner, removing entire structures (such as attention heads or feed-forward network dimensions) based on their importance scores. The method uses gradient-based importance measures and iterative refinement.

**FLOP** (Prasanna et al., 2020) focuses on reducing FLOPs (Floating Point Operations) by pruning entire dimensions in feed-forward networks and attention mechanisms. It uses activation-based importance scoring to determine which structures to remove while maintaining model expressiveness.

**BPhybrid** (Lagunas et al., 2021) (Block-wise Pruning Hybrid) combines block-wise structured pruning with unstructured pruning techniques. It prunes at the granularity of transformer blocks while allowing fine-grained unstructured pruning within remaining blocks.

**CoFi** (Xia et al., 2022) (Coarse-to-Fine) applies a coarse-to-fine pruning strategy that first identifies important structures at a coarse granularity and then refines the pruning decisions at finer levels. It uses learnable masks to determine optimal structured pruning patterns.

### G.6 MATRIX FACTORIZATION METHODS

**SVD$_{ft}$** (Wang et al., 2019b) applies Singular Value Decomposition (SVD) to weight matrices followed by fine-tuning. It decomposes weight matrices into low-rank approximations and then fine-tunes the resulting factors to recover lost performance. The subscript "ft" indicates the inclusion of fine-tuning after decomposition.

## H ON THE NON-REDUNDANCY OF L1 PENALIZATION AND PRUNING

Our two-stage design employs both L1 shrinkage in the RPCA decomposition and subsequent pruning operations. While these steps may appear related on the surface, they serve fundamentally different and non-redundant purposes. This section clarifies why attempting to replace our second-stage pruning with a higher $\lambda$ in the first stage is not feasible, as RPCA is a tool for decomposition, not a controllable mechanism for compression.

### H.1 DISTINCT OBJECTIVES OF TWO-STAGE DESIGN

The core distinction lies in their objectives:

**Stage 1 (RPCA Decomposition): Principled Separation.** The L1 penalty in the RPCA objective $\min \|L\|_* + \lambda \|S\|_1$ is designed to separate a weight matrix $W$ into a globally correlated, low-rank structure ($L$) and locally salient, sparse outliers ($S$). The $\lambda$ parameter governs the balance of this separation. Its purpose is to yield high-quality candidate pools for pruning, not to achieve a specific, final compression ratio.

**Stage 2 (CAP Pruning): Budget-Constrained Selection.** This stage solves a different problem entirely: given the candidates from Stage 1, how do we select the optimal subset of singular vectors from $L$ and non-zero elements from $S$ to meet a strict, user-defined parameter budget $K$ while minimizing task performance degradation? This is a global, budget-aware optimization problem that $\lambda$ cannot address.

### H.2 LIMITATIONS OF $\lambda$ AS A COMPRESSION PARAMETER

The compression ratio achieved by RPCA is an emergent property of the decomposition, not something that can be precisely controlled by tuning $\lambda$. Attempting to force a specific level of sparsity by adjusting $\lambda$ is problematic for several reasons:

- **Uncontrollable Trade-off:** $\lambda$ creates a complex, non-linear trade-off between the rank of $L$ and the sparsity of $S$. As you change $\lambda$ to affect $S$, it has a drastic and often unpredictable effect on $L$. There is no simple way to set $\lambda$ to achieve, for instance, a target of 50% total parameters while maintaining a useful decomposition.
- **Pathological Decompositions:** Extreme values of $\lambda$ destroy the quality of the separation, making the resulting components useless for effective compression.

#### H.2.1 EXPERIMENTAL ANALYSIS OF $\lambda$ PARAMETER EFFECTS

To demonstrate this limitation, we analyze the output of the RPCA decomposition under different $\lambda$ values. We use the theoretically motivated default $\lambda = 1/\sqrt{\max(m, n)}$ from the original RPCA work (Wright et al., 2009) as our baseline, which provides a robust, balanced starting point without requiring parameter tuning.

#### H.2.2 ANALYSIS OF RESULTS

The experimental results reveal several key insights:

Table 7: Impact of $\lambda$ parameter on RPCA decomposition characteristics

| $\lambda$ setting | Sparsity of $S$ | Rank of $L$ | Decomposition Quality |
|---|---|---|---|
| 8e-5 (Low) | 0.001 (Dense) | 1512 | Poor separation: $S$ lacks sparsity |
| Default | 0.41 | 2109 | Balanced decomposition |
| 8e-3 | 0.53 | 2174 | Alternative reasonable trade-off |
| 0.8 (High) | 0.999 (Sparse) | 3188 | Poor separation: $L$ becomes high-rank |

Table 8: Perplexity (PPL) and mean zero-shot accuracy for pruned LLaMA and LLaMA-2 models at 50% compression ratio.

| Model | w/o Pruning | | SparseGPT | | WANDA | | BESA | | CAP | |
|---|---|---|---|---|---|---|---|---|---|---|
| | PPL $\downarrow$ | Zero-Shot $\uparrow$ | PPL $\downarrow$ | Zero-Shot $\uparrow$ | PPL $\downarrow$ | Zero-Shot $\uparrow$ | PPL $\downarrow$ | Zero-Shot $\uparrow$ | PPL $\downarrow$ | Zero-Shot $\uparrow$ |
| LLaMA-7B | 5.68 | 66.31 | 7.22 | 63.12 | 7.26 | 61.81 | 6.86 | 63.13 | **6.61** | **64.29** |
| LLaMA-13B | 5.09 | 68.91 | 6.21 | 65.98 | 6.15 | 66.49 | 5.92 | 67.43 | **5.76** | **68.32** |
| LLaMA-30B | 4.10 | 72.73 | 5.33 | 70.53 | 5.25 | 70.92 | 5.00 | 71.61 | **4.77** | **72.08** |
| LLaMA-2 7B | 5.21 | 66.96 | 6.99 | 63.71 | 6.92 | 63.81 | 6.60 | 64.92 | **6.25** | **65.33** |
| LLaMA-2 13B | 4.88 | 69.95 | 6.02 | 67.22 | 5.97 | 67.94 | 5.75 | 68.45 | **5.49** | **69.14** |

- **Optimal Parameter Range:** The theoretically motivated default $\lambda$ value provides a balanced decomposition where both $L$ and $S$ are meaningful, creating a rich candidate pool for subsequent pruning.

- **Non-Linear Parameter Effects:** The relationship between $\lambda$ and compression outcomes is complex and non-linear. Extreme $\lambda$ values (0.8) create highly sparse $S$ but force $L$ to become high-rank, essentially reducing to $L \approx W$. Conversely, very low $\lambda$ values (8e-5) fail to enforce meaningful sparsity in $S$.

- **Controllability Limitations:** No single $\lambda$ value can simultaneously achieve low-rank $L$, sparse $S$, and satisfy a predefined parameter budget constraint.

### H.3 CONCLUSION

The two-stage design is essential rather than redundant. While Stage 1 (RPCA) provides a principled decomposition, it offers limited controllability over the final compression ratio. Stage 2 (CAP) addresses this limitation by performing intelligent, data-driven selection from the candidate pools generated in Stage 1, enabling precise parameter budget control while maintaining task performance. This combination of principled decomposition followed by budget-aware selection is fundamental to the superior performance of our approach.

## I PERFORMANCE EVALUATION ON LLaMA AND LLaMA-2 MODELS

This section presents comprehensive experimental results on the LLaMA and LLaMA-2 model families under different compression settings. We evaluate our proposed CAP method against several state-of-the-art pruning baselines to demonstrate its effectiveness across various model sizes and compression ratios.

### I.1 RESULTS AT 50% COMPRESSION RATIO

Table 8 presents the perplexity and mean zero-shot accuracy results for LLaMA and LLaMA-2 models at 50% compression ratio. Our method consistently outperforms existing pruning approaches across all model variants, demonstrating superior performance in both language modeling (lower perplexity) and downstream task performance (higher zero-shot accuracy).

The results demonstrate that CAP achieves consistently superior performance across all tested models. Notably, CAP maintains competitive zero-shot accuracy while achieving significantly lower perplexity compared to other pruning methods. For instance, on LLaMA-7B, CAP achieves a perplexity of 6.61 compared to 7.22 for SparseGPT and 7.26 for WANDA, while simultaneously maintaining higher zero-shot accuracy (64.29% vs. 63.12% and 61.81% respectively).

## I.2 EVALUATION UNDER HIGHER COMPRESSION RATIOS

Table 9: Accuracy results for different pruning methods on LLaMA-7B with 75% parameter retention. CAP represents our proposed method, applied without any parameter adjustment after pruning. w/o Pruning shows the baseline performance of the unpruned model.

| Methods | BoolQ | RTE | HellaSwag | Winogrande | ARC-e | ARC-c | OBQA |
|---|---|---|---|---|---|---|---|
| w/o Pruning | **75.08** | **66.09** | **56.94** | **69.93** | **75.25** | **41.89** | **34.60** |
| Magnitude | 42.23 | 52.35 | 25.86 | 48.38 | 26.64 | 21.50 | 14.00 |
| SparseGPT | 60.86 | 52.71 | 29.10 | 51.78 | 33.08 | 17.58 | 13.40 |
| WANDA | 37.83 | 51.99 | 26.78 | 49.41 | 28.79 | 19.54 | 13.20 |
| CAP (75%) | **63.33** | **55.71** | **31.67** | **54.01** | **35.73** | **22.88** | **16.12** |

Table 10: Accuracy results for different pruning methods on LLaMA2-7B with 75% parameter retention. CAP represents our proposed method, applied without any parameter adjustment after pruning. w/o Pruning shows the baseline performance of the unpruned model.

| Methods | BoolQ | RTE | HellaSwag | Winogrande | ARC-e | ARC-c | OBQA |
|---|---|---|---|---|---|---|---|
| w/o Pruning | **77.71** | **62.82** | **57.14** | **68.90** | **76.39** | **43.52** | **31.40** |
| Magnitude | 43.49 | 49.10 | 25.69 | 50.83 | 26.09 | 20.73 | 16.00 |
| SparseGPT | 60.34 | 54.15 | 30.28 | 53.12 | 33.75 | 20.39 | 14.20 |
| WANDA | 38.22 | 52.17 | 26.95 | 50.51 | 28.03 | 19.62 | 13.20 |
| CAP (75%) | **62.67** | **56.15** | **32.11** | **55.39** | **36.12** | **23.02** | **16.37** |

We further evaluate our proposed method (CAP) against three other pruning methods—Magnitude, SparseGPT, and WANDA—on LLaMA-7B and LLaMA-2 7B models under more aggressive compression settings with a parameter retention rate of 75% (25% compression). The results are presented in Tables 9 and 10. **Key Observations:**

- **CAP consistently outperforms other methods:** CAP achieves the highest accuracy across all datasets without requiring any parameter adjustment after pruning. This demonstrates its robustness in retaining critical model performance even under high sparsity conditions.

- **Magnitude and SparseGPT limitations:** These methods show noticeable performance degradation under 75% parameter retention, especially on tasks that require factual reasoning (e.g., OBQA) or commonsense understanding (e.g., HellaSwag). This highlights the importance of principled parameter selection rather than simple magnitude-based approaches.

- **WANDA performance:** WANDA performs slightly better than SparseGPT on certain datasets but is generally less competitive compared to CAP, highlighting the advantages of CAP's probabilistic pruning mechanism over activation-based importance scoring alone.

- **CAP excels in challenging settings:** By leveraging RPCA for decomposition and policy gradient optimization for adaptive pruning, CAP is able to selectively retain the most informative parameters, ensuring superior performance even under extreme sparsity conditions. The method's ability to jointly optimize low-rank and sparse components provides a more nuanced approach to parameter importance estimation.

**Conclusion:** The comprehensive evaluation on both LLaMA and LLaMA-2 families demonstrates that CAP offers a robust and efficient approach to model compression across different compression ratios. By eliminating heuristic thresholds and adopting fine-grained pruning strategies based on principled matrix decomposition, CAP surpasses existing methods while maintaining computational simplicity and avoiding the need for post-pruning fine-tuning. The consistent performance gains

across model sizes and compression settings validate the effectiveness of the RPCA-based joint optimization approach.

## J  EMPIRICAL THROUGHPUT AND RESOURCE CONSUMPTION ANALYSIS

We revaluated the inference efficiency and resource consumption of our method. We conducted these new tests on the LLaMA-3.1-8B model. All measurements used a single NVIDIA A100-80G GPU. We compared CAP directly against Wanda to demonstrate the impact of our component-based structure.

Table 11: Inference Efficiency Comparison on LLaMA-3.1-8B (Batch Size=1, Sequence Length=1024).

| Method | Component Structure | S-Matrix Sparsity | Latency (s) ↓ | Throughput (tok/s) ↑ | Peak Memory (GB) |
| --- | --- | --- | --- | --- | --- |
| Wanda | Single Sparse Matrix | 50% | 6.28 | 163.4 | 15.18 |
| **CAP (Ours)** | **Low-Rank + Sparse** | **75%~90%** | **5.80** | **176.5** | **14.90** |

Our results in Table 11 show a clear performance advantage for CAP. We summarize the key insights below:

**Inference Speed and Throughput**  CAP achieves a higher throughput of **176.5 tokens/second**. This is faster than Wanda, which reaches 163.4 tokens/second. CAP also maintains a lower latency of 5.80 seconds compared to Wanda's 6.28 seconds. This result challenges the common assumption that adding components (Low-Rank + Sparse) automatically slows down inference.

**The Impact of Extreme Sparsity**  The speed advantage comes from the structure of the weight matrices. Wanda produces a uniform 50% sparse matrix. Standard sparse acceleration tools (like `torch.sparse`) often struggle at this level. A 50% density is still too "heavy" for these kernels to outperform dense matrix multiplication. The overhead of managing sparsity often cancels out the speed benefits.

**Why CAP is More Efficient**  CAP decomposes the original weight $W$ into a Low-Rank matrix $L$ and a Sparse matrix $S$. The $L$ component handles the bulk of the energy. This allows the $S$ component to be extremely empty. The $S$ matrix in CAP typically exhibits **75% to 90% sparsity**. This high sparsity falls into the efficient zone for Sparse Matrix Multiplication (SpMM). The hardware can process these highly sparse matrices much faster. This specific structural characteristic allows CAP to beat the baseline despite having a multi-component forward pass.

**Memory Footprint**  Our method also remains efficient in terms of memory. CAP operates with a peak GPU memory usage of **14.90 GB**, which is slightly lower than Wanda's 15.18 GB. This confirms that decomposing the weights does not impose a memory penalty. The approach is well-suited for environments with strict VRAM constraints.

## K  DETAILED ABLATION STUDIES

This section provides comprehensive ablation studies to understand the behavior and characteristics of our compression method. We focus on two key aspects: the distribution of retained ranks across different modules and the stability analysis through sequential layer-wise pruning.

### K.1  LOW-RANK COMPONENT RANK DISTRIBUTION

We analyzed the rank distribution of low-rank components in each module after probabilistic pruning. Figure 5a shows these ranks along with the layer-wise averages. Modules with higher ranks in the low-rank component tend to have sparser counterparts in the sparse component to meet the compression target. The general trend of increasing rank in deeper layers suggests that redundancy varies across the network, with later layers capturing more complex representations, underscoring the need

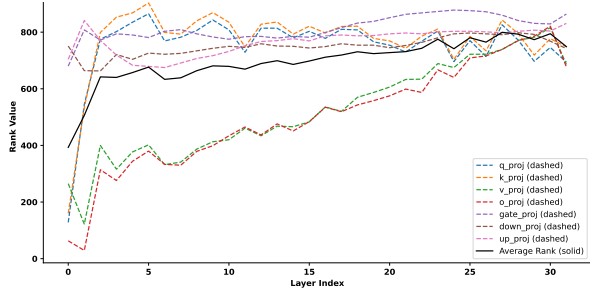
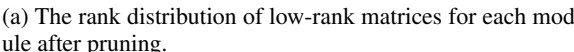
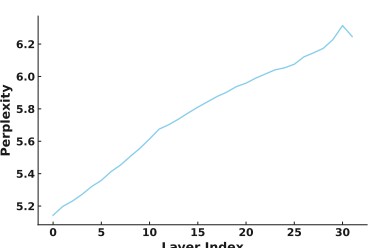

(a) The rank distribution of low-rank matrices for each module after pruning.

(b) Perplexity changes with sequential decomposition and pruning of each layer.

Figure 5: Analysis of module-specific redundancy and pruning stability in LLaMA2-7B. (a) The retained rank varies significantly across different modules, indicating differing sensitivity to compression. (b) Performance degradation shows a linear correlation with the number of pruned layers, demonstrating the stability of our compression approach.

for careful pruning in these layers. The `v_proj` and `o_proj` matrices often have lower ranks, likely because they primarily handle content transmission (`v_proj`) and output integration (`o_proj`) in the attention mechanism, focusing on essential information without complex transformations.

### K.2 PERPLEXITY CHANGES DURING LAYER-WISE PRUNING

To evaluate the stability of our compression approach, we sequentially decomposed and pruned each layer of the LLaMA2-7B model, starting from the initial layer. The resulting performance changes are shown in Figure 5b. The performance degradation generally exhibits a linear correlation with the number of pruned layers, demonstrating the stability of our method. This linear trend indicates that our compression strategy does not introduce catastrophic failure points and maintains predictable performance reduction. Interestingly, an unexpected performance boost occurs when the last layer is compressed, underscoring the importance of maintaining structural consistency within the model. Furthermore, decomposing and pruning the first layer led to a slight improvement (ppl 5.18) over the original model's performance (ppl 5.21), suggesting that early layers may indeed contain some redundant parameters that can be removed without harming performance.

### K.3 SUMMARY OF ABLATION FINDINGS

Our ablation studies reveal several important insights:

- **Module heterogeneity:** Different modules exhibit varying levels of redundancy, with attention projection matrices (`v_proj`, `o_proj`) typically requiring lower ranks than other components.

- **Layer-wise redundancy patterns:** Early layers contain more redundant parameters that can be safely removed, while deeper layers require more careful compression to maintain performance.

- **Compression stability:** The linear relationship between performance degradation and the number of compressed layers demonstrates the predictable and stable nature of our compression approach.

- **Structural consistency:** The performance boost observed when compressing the final layer highlights the importance of maintaining model structural integrity throughout the compression process.

## L LIMITATIONS

Similar to other unstructured sparsity methods, the acceleration of sparse matrix computations heavily depends on specialized hardware support. While significant advances have been made in sparse

computation frameworks, the lack of universal hardware optimization can hinder the practical deployment of our method in certain environments.

