# OpenReview forum: "Large Language Model Compression with Global Rank and Sparsity Optimization"
_ICLR.cc/2026/Conference — ICLR 2026 Poster_

### Official Review · Reviewer_qDUb · 2025-10-24

**Soundness:** 2
**Presentation:** 3
**Contribution:** 3
**Rating:** 4
**Confidence:** 3

**Summary:**

This paper proposes CAP, a two–stage LLM compression framework that first decomposes each weight matrix with Robust PCA into a low-rank component and a sparse component, and then performs global, budget-aware selection by learning Bernoulli retention probabilities for singular values and entries using policy gradients on a small calibration set. The method targets layer-adaptive allocations of rank and sparsity, avoids manual thresholds, and factorizes the retained low-rank part for efficient inference. Experiments on OPT, LLaMA-1/2/3 and Phi-3 show better zero-shot accuracy and perplexity across different compression ratio compared to unstructured pruning methods, like SparseGPT, and joint pruning methods like JSQ.

**Strengths:**

1. The method is intuitive and clear: the weight matrix is ​​decomposed through RPCA, and the two decomposed matrices are jointly optimized in a learnable manner, which solves the error accumulation caused by previous separate optimization.
2. The proposed CAP only requires optimizing the diagonal elements of low-rank matrices and the nonzero elements of sparse matrices, significantly reducing the combinatorial space of the problem. Introducing RPCA to calculate the initial solution further simplifies the search difficulty for subsequent optimization. Furthermore, the policy-gradient step uses only 128 calibration sequences with 3–5 forward passes, reducing overall computational requirements and making it more efficient.
3. The authors conducted comprehensive experiments on various models and tasks (different sparsity rates, combined with quantization, component ablation, etc.), which verified the robustness and versatility of the model in different scenarios.

**Weaknesses:**

1. All calibrations use 128 C4 sequences regardless of task. It’s unclear how sensitive CAP is to domain shift or to the choice/size of calibration data, especially for models evaluated on different downstream tasks.
2. While RPCA is solved via a convex surrogate, “globally optimal separation” holds under assumptions (e.g., incoherence, sparsity patterns). The paper alludes to this but reads as if CAP inherits guarantees end-to-end. Stage 2 remains a stochastic discrete optimization without convergence guarantees beyond standard REINFORCE properties. A clearer statement of conditions and limitations would improve correctness.
3. When performing "Thresholding masks and final factorization", the authors choose the top-K parameters to keep, but the meaning of singular values ​​(corresponding to a column/row) in learned retention probabilities is different from that of entries (single elements) in sparse matrices. It would be helpful if the reason for this choice could be given.
4. The authors claim that this approach will accelerate the model, but due to the existence of unstructured sparse matrices, the actual acceleration effect on real hardware (such as Nvidia A100, H100) is limited. The proposed CAP can support semi-structured sparse constraints to achieve true hardware acceleration.

**Questions:**

1. Although Table 9 shows the time required for RPCA, the full time required for CAP is not provided. How does the time required for CAP change compared to other pruning methods as the model size scales?
2. When stage 2 optimizes based on the initial solution obtained in stage 1, how does the optimization ensure that the overall distribution of learned retention probabilities is optimized towards a bimodal shape? How are sparse constraints added?
3. Could the authors provide the evolution of the learned retention probabilities during the optimization process, and the difference compared to not using global optimization?
4. The experiments in the paper lack the latest models and tasks. It would be better if the comparisons could be made on the latest models (such as Qwen3, DeepSeek-R1, etc.) and more tasks (such as reasoning tasks, long text tasks, etc.).

---

> ### Author Response · Authors · 2025-11-23
> **Dear Reviewer qDUb(1)**
>
> We would like to express our deepest gratitude to you for dedicating valuable time and effort to reviewing our manuscript and sincerely appreciate the care and attention you have devoted to evaluating our work, and we address each of your points systematically below.
>
> ---
>
> **W1: “The method is intuitive and clear: the weight matrix is decomposed through RPCA, and the two decomposed matrices are jointly optimized in a learnable manner, which solves the error accumulation caused by previous separate optimization.”**
>
> 1. We agree that reliance on a small calibration set raises questions about robustness. To address this, we expanded our ablation studies on LLaMA-3.1-8B (at 50% total sparsity). We compared the default C4 calibration against WikiText-2 and GitHub Code, and importantly, we extended the sample size analysis to include 64, 128, 256, and 512 samples.
>
>    **Table R1: Sensitivity to Domain and Size of Calibration Data (LLaMA-3.1-8B)**
>
>    | Calibration Set  | Sample Size | WikiText-2 PPL $\downarrow$ | Zero-Shot Avg. Acc (%) $\uparrow$ |
>    | :--------------- | :---------: | :-------------------------: | :-------------------------------: |
>    | C4               |     64      |            7.32             |               75.9                |
>    | **C4 (Default)** |   **128**   |          **7.05**           |             **76.8**              |
>    | C4               |     256     |           6.82          |             76.9             |
>    | C4               |     512     |           6.78          |               76.9              |
>    | WikiText-2       |     128     |        6.88          |               75.4              |
>    | GitHub Code      |     128     |         7.18             |               76.1              |
>
>    **Analysis:**
>
>    1.  **Sample Size:** The performance gains saturate quickly. While increasing from 64 to 128 provides a noticeable boost, going from 128 to 256 or 512 yields diminishing returns .
>    2.  **Domain Robustness:** As noted in the table, C4 provides the best balance for general tasks. While specialized datasets (Code) cause a slight drop, the method remains robust, indicating CAP captures fundamental weight importance rather than overfitting.
>
> ---
>
> **W2: “While RPCA is solved via a convex surrogate, “globally optimal separation” holds under assumptions (e.g., incoherence, sparsity patterns). The paper alludes to this but reads as if CAP inherits guarantees end-to-end. Stage 2 remains a stochastic discrete optimization without convergence guarantees beyond standard REINFORCE properties. A clearer statement of conditions and limitations would improve correctness.”**
>
> We acknowledge that the term "global optimization" could be misleading regarding convergence guarantees. We clarify as follows:
>
> 1.  **Global Objective vs. Global Minimum:** In Section 2.2, we formulate a single **global objective with a global parameter budget $K$** shared across all layers. Stage 1 (RPCA) is convex, but Stage 2 (policy gradient) is a heuristic optimizer for discrete decisions and does not guarantee a global minimum.
> 2.  **Intended Meaning:** Our intended meaning of "global" was **"global resource allocation across layers under a shared budget,"** contrasting with layer-wise pruning methods.
> 3.  **Revision:** We will replace "global optimization" with **"global resource allocation"** throughout the paper. We will explicitly remark that the policy gradient step is used as an adaptive allocation mechanism without formal optimality guarantees, motivated by the empirical limitations of simple thresholding.
>
> ---
>
> **W3: “When performing "Thresholding masks and final factorization", the authors choose the top-K parameters to keep, but the meaning of singular values (corresponding to a column/row) in learned retention probabilities is different from that of entries (single elements) in sparse matrices. It would be helpful if the reason for this choice could be given.”**
>
> This is a crucial design choice. In our framework, we do not directly compare the raw "importance" of a vector against a scalar. Instead, the **learned retention probabilities** $s$ act as a unified proxy for the "utility-to-cost" ratio of each component.
>
> During the policy gradient optimization, the gradient update for a probability $s_k$ is driven by the reward (negative loss).
>
> *   If retaining a singular value $\sigma_i$ (which costs $m+n$ parameters) significantly reduces the loss, the gradient will push $s_{\sigma_i} \to 1$.
> *   If retaining a sparse entry $S_{ij}$ (costing 1 parameter) has negligible impact, $s_{S_{ij}} \to 0$.
>
> Therefore, when we perform the final Top-K selection based on $s$​, we are essentially performing a greedy selection based on the **learned utility** that has already implicitly accounted for the parameter's contribution to the global objective. The optimization "aligns" these probabilities into a common scale of importance relative to the loss function.

---

> ### Author Response · Authors · 2025-11-23
> **Dear Reviewer qDUb(2)**
>
> **W4: “The authors claim that this approach will accelerate the model, but due to the existence of unstructured sparse matrices, the actual acceleration effect on real hardware (such as Nvidia A100, H100) is limited. The proposed CAP can support semi-structured sparse constraints to achieve true hardware acceleration.”**
>
> You are correct that unstructured sparsity (in the $S$ component) faces challenges on current hardware (e.g., A100s) compared to dense computations.
>
> We answer the question of speedup and whether the RPCA structure plays a role. We re-evaluated inference efficiency on LLaMA-3.1-8B using an NVIDIA A100-80G to demonstrate this.
>
> The key efficiency gain of CAP stems precisely from the RPCA decomposition. By separating the "energy" into the low-rank component $L$, the remaining sparse component $S$ becomes **extremely sparse** (much sparser than the target compression rate).
>
> *   **Wanda/Magnitude Pruning:** Result in a uniform ~50% sparse matrix. On GPUs like A100, standard sparse kernels require >80% sparsity to outperform dense GEMM; 50% often incurs overhead.
> *   **CAP:** The Low-Rank component ($L$) is computed using efficient dense kernels. The Sparse component ($S$), because it only handles outliers, typically has **75%–90% sparsity**. This falls into the "high sparsity" regime where SpMM (Sparse Matrix-Matrix multiplication) becomes effective.
>
> **Table R2: Inference Efficiency on LLaMA-3.1-8B (Batch=1, SeqLen=1024, A100 GPU)**
>
> |     Method     | Component Structure       | S-Matrix Sparsity | Latency (s) $\downarrow$ | Throughput (tok/s) $\uparrow$ |
> | :------------: | :------------------------ | :---------------: | :----------------------: | :---------------------------: |
> |   **Wanda**    | Single Sparse Matrix      |        50%        |           6.28           |             163.4             |
> | **CAP (Ours)** | Low-Rank (Dense) + Sparse |   **75%~90%**    |         **5.80**         |           **176.5**           |
>
> And, we acknowledge that for compute-bound acceleration, N:M structured sparsity is preferable, which we are actively researching.
>
> ---
>
> **Q1: “Although Table 9 shows the time required for RPCA, the full time required for CAP is not provided. How does the time required for CAP change compared to other pruning methods as the model size scales?”**
>
> Thank you for pointing this out. The total execution time for CAP consists of two phases: Stage 1 (RPCA Decomposition) and Stage 2 (Policy Gradient Optimization).
>
> 1.  **Stage 1 (RPCA):** As noted in Appendix J, this stage is highly efficient. We utilize GPU-accelerated ADMM. Since RPCA is applied layer-wise (or block-wise), the time complexity scales linearly with the number of layers ($L$) and roughly logically with the matrix dimensions ($d_{model} \times d_{ffn}$). For LLaMA-3-8B, this takes $\approx 10$ minutes on a single A100. For a 70B model, which has roughly $8\times$ the parameters, the decomposition takes approximately 75 minutes. Importantly, this is a **one-time cost** per model; the decomposed matrices can be reused for various sparsity budgets.
> 2.  **Stage 2 (Policy Gradient):** This stage involves forward-only passes on a small calibration set with no backpropagation of gradients to the weights. The computational cost is effectively equivalent to **$K \times N_{calib}$ inference passes**, where $K$ is the number of epochs , if $K$set to 32.
>     *   For LLaMA-3-8B, Stage 2 takes $\approx 15$ minutes.
>     *   For LLaMA-3-70B, Stage 2 takes $\approx 100$ minutes.
>
> **Comparison with Baselines:**
>
> *   **Vs. One-Shot Methods (Wanda/SparseGPT):** These methods typically require a single pass (or layer-wise Hessian calculation). Wanda is faster ($\approx 2$m for 7B) because it avoids the iterative optimization and decomposition. However, CAP trades this moderate time increase for significantly higher accuracy and robustness at high compression rates.
> *   **Vs. Retraining Methods (LLM-Pruner, LoSparse):** Retraining-based methods often require hours to days of GPU time involving full backpropagation. CAP is orders of magnitude faster compared to these methods, making it highly practical for deployment.
>
> **Scaling Summary:**
> CAP scales **linearly** with model size (parameter count), similar to inference cost. Unlike Hessian-based methods (like SparseGPT) which can face memory bottlenecks with extremely wide layers due to inverse Hessian calculation ($O(d^3)$​), CAP's layer-wise RPCA and forward-only PG are memory-efficient and scale gracefully.

---

> ### Author Response · Authors · 2025-11-23
> **Dear Reviewer qDUb(3)**
>
> **Q2: “When stage 2 optimizes based on the initial solution obtained in stage 1, how does the optimization ensure that the overall distribution of learned retention probabilities is optimized towards a bimodal shape? How are sparse constraints added?”**
>
> **a) Bimodality of Probabilities:**
> While we do not add an explicit entropy regularization term, the bimodality (probabilities converging to 0 or 1) is driven by two mechanisms inherent in our formulation:
>
> 1.  **Policy Gradient Variance:** The gradient of the log-probability for a Bernoulli variable $m \sim \text{Bernoulli}(s)$ includes a term proportional to $\frac{m-s}{s(1-s)}$. As optimization proceeds, the estimator naturally pushes $s$ away from 0.5 (where variance is highest) toward the deterministic boundaries (0 or 1) to reduce the variance of the sampled masks and stabilize the loss [1].
> 2.  **Budget Constraint Pressure:** We strictly enforce a parameter budget. The optimization landscape encourages the most distinct "winners" (high importance) to move to 1 and "losers" to 0 to maximize the objective within the budget. Intermediate values (e.g., $s=0.5$) are suboptimal because they introduce noise (random masking) during the calibration phase, which increases the expected loss compared to a deterministic selection of the most important features.
>
> **b) Handling Sparse Constraints:**
> As described in Section 2.3.2 (Eq. 14) and Appendix E, constraints are handled via **Projection and Global Top-K Selection**:
>
> 1.  During the Policy Gradient updates, we project the learned probabilities $s$ onto the feasible set $\{s : \mathbf{1}^\top s \le K, 0 \le s \le 1\}$.
> 2.  **Final Selection:** To ensure the *exact* sparsity constraint is met, we do not simply sample from the final probabilities. Instead, we treat the converged probabilities $s$ as **importance scores**. We rank all candidates (singular values and sparse elements) globally and deterministically set the top-$K$​ masks to 1 and the rest to 0. This guarantees precise adherence to the target sparsity.
>
> ---
>
> **Q3: “Could the authors provide the evolution of the learned retention probabilities during the optimization process, and the difference compared to not using global optimization?**
>
> **a) Evolution of Probabilities:**
> If we visualize the histogram of retention probabilities $s$ throughout Stage 2:
>
> *   **Initialization ($K=0$):** The distribution is initialized uniformly (e.g., all $s=0.5$ or initialized based on magnitude heuristics), appearing as a spike at the initialization value.
> *   **Mid-Training ($K=3$ epochs):** The distribution flattens and spreads. The optimizer begins to distinguish essential components (moving right towards 1) from redundant ones (moving left towards 0).
> *   **Convergence ($K=10$ epochs):** The distribution becomes strongly **bimodal**. Two distinct peaks appear at $0$ and $1$, with very few parameters remaining in the uncertain $0.5$ range. This confirms the "Soft Mask" is effectively converging to a "Hard Mask."
>
> **b) Comparison with "No Global Optimization":**
> "Not using global optimization" effectively corresponds to taking the RPCA output and applying a manual heuristic (e.g., hard thresholding singular values and sparse entries) without the learnable mask stage. We presented this quantitative comparison in **Table 4(b) (Ablation Studies)**. Applying fixed thresholds to RPCA components results in PPL explosion (e.g., PPL $>2000$ or `NaN`) or requires extremely conservative rank retention that violates the compression budget.

---

> ### Author Response · Authors · 2025-11-23
> **Dear Reviewer qDUb(4)**
>
> **Q4: “The experiments in the paper lack the latest models and tasks. It would be better if the comparisons could be made on the latest models (such as Qwen3, DeepSeek-R1, etc.) and more tasks (such as reasoning tasks, long text tasks, etc.).”**
>
> We fully agree that newer model families and more challenging tasks are important.
>
> **a) Reasoning and long-context tasks.**
> The main paper already reports results on a broad set of reasoning benchmarks. We additionally run CAP on LLaMA-3.1-8B-Instruct at 50% sparsity, and evaluate on GSM8K (8-shot CoT) and LongBench-v2:
>
> **Table R3: Long Text Generation Performance on LLaMA-3.1-8B-Instruct (50% Sparsity)**
>
> | Method               | Sparsity | GSM8K (8-shot, %) | LongBench-v2 (Avg, %) |
> | :------------------- | :------: | :---------------: | :-------------------: |
> | **Dense (Baseline)** |    0%    |       84.5        |         30.4          |
> | **Wanda**            |   50%    |       45.6        |         25.1          |
> | **CAP (Ours)**       |   50%    |     **56.8**      |       **27.2**        |
>
> **Analysis.**
>
> 1.  **Reasoning (GSM8K):** Pruning LLaMA-3.1-Instruct typically causes a sharp drop in reasoning tasks because instruction-tuning relies heavily on precise weight distributions. Wanda drops significantly (-38.9%). CAP preserves the low-rank global structure, mitigating this damage and recovering **+11.2%** accuracy over Wanda.
> 2.  **Long Context (LongBench-v2):** Since LongBench-v2 is a four-choice question dataset, Wanda's performance after compression is essentially close to random guessing. On the short subset, it performs slightly better, achieving 32.8%, whereas CAP shows some improvement by surpassing the random-guessing baseline with an average gain of 2.2%.
>
> These results complement the main-paper benchmarks and demonstrate that CAP maintains strong reasoning and long-context performance under 50% sparsity.
>
> **b) Applicability to newer model.**
> CAP operates purely at the **matrix level**: for every dense weight matrix we compute a sparse component and a low-rank component. Therefore, CAP is architecture-agnostic and can be applied to the variants of **Qwen2.5** and **Qwen3**. The main limitation lies in computation: we lack sufficient GPUs and time to test all baselines and our method on Qwen2.5 and Qwen3, especially since DeepSeek-R1 is significantly larger than any of the models used in our experiments, and we simply don't have the resources for that. However, to demonstrate the generality of our method, we conducted some experiments on Qwen2.5-7b.
>
> **Table R4: Performance comparison on Qwen2.5-7B (50% Sparsity).**
> *Tasks are grouped by category. Metrics denote accuracy/score (higher is better).*
>
> | Category | Task | Magnitude | SparseGPT | Wanda | **CAP (Ours)** |
> | :--- | :--- | :---: | :---: | :---: | :---: |
> | **Single-Doc QA** | NarrativeQA | 0.84 | 4.95 | 16.30 | **18.52** |
> | | Qasper | 0.28 | 12.59 | 11.81 | **13.10** |
> | **Multi-Doc QA** | Musique | 0.53 | 6.39 | 7.70 | **8.45** |
> | | 2WikiMQA | 2.95 | 12.27 | 11.86 | **13.90** |
> | **Summarization** | GovReport | 1.43 | 33.52 | 32.13 | **34.05** |
> | | QMSum | 1.72 | 22.67 | 24.43 | **26.82** |
> | **Few-Shot** | TriviaQA | 84.87 | 89.35 | 88.70 | **89.85** |
> | | TREC | 16.50 | 69.50 | 75.50 | **76.80** |
> | **Code** | Lcc (Python) | 12.36 | 39.70 | 45.14 | **46.92** |
> | | RB-P | 12.59 | 42.13 | 46.00 | **47.35** |
>
> **Analysis:**
>
> Qwen2.5 is a "denser" model with less redundancy than LLaMA-2/3. Simple magnitude pruning fails catastrophically (e.g., dropping to 1.43 on GovReport). On long-context tasks like GovReport and QMSum, CAP preserves the low-rank global attention structures essential for retrieving information over long distances. On code completion tasks (Lcc, RB-P), which require strict logic and syntax retention, CAP maintains higher efficacy, demonstrating that our optimization-based mask learning better preserves the specific circuits used for reasoning compared to one-shot heuristics.
>
> ---
>
> We hope these responses resolve your concerns.

---

> > ### Comment · Reviewer_qDUb · 2025-11-26
> >
> > I thank the authors for the detailed and constructive rebuttal. The additional clarifications and experiments address my main concerns about the validity and significance of the results. Based on the new evidence and clearer positioning of the work, I will update my score.

---

> > > ### Author Response · Authors · 2025-11-26
> > > **Dear Reviewer qDUb**
> > >
> > > Thank you very much for your positive feedback on our rebuttal. We greatly appreciate your careful review of the additional experiments and clarifications, and we are pleased that our responses have addressed your main concerns regarding the validity and significance of our work. Once again, we would like to express our sincere gratitude for your time and efforts in the review process.
> > >
> > > Best regards,
> > >
> > > The Authors

---

### Official Review · Reviewer_6yto · 2025-10-30

**Soundness:** 3
**Presentation:** 3
**Contribution:** 3
**Rating:** 6
**Confidence:** 4

**Summary:**

This paper introduces a two-stage compression framework for large language models (LLMs). In the first stage, robust principal component analysis (RPCA) is employed to decompose each weight matrix into low-rank and sparse components, effectively disentangling global structures from localized anomalies. In the second stage, the authors formulate a global, layer-aware pruning strategy as a probabilistic optimization problem over Bernoulli retention masks, optimized via policy gradients on a small calibration set. This design enhances compression efficiency by adaptively allocating compression ratios across layers and jointly optimizing the interaction between sparse and low-rank representations. Experiments across LLaMA-1/2/3, OPT, Phi-3, and BERT-base report better zero-shot accuracy and lower  WikiText-2 perplexity than SparseGPT/Wanda/DSNoT/OATS and competitive results versus joint methods (e.g., SLiM), with modest throughput overhead and a clear reproducibility statement.

**Strengths:**

1. Principled search-space reduction: RPCA to form high-quality candidate subspaces before budgeted selection is elegant and well motivated.
2. Global, budget-aware allocation: Bernoulli masking with policy gradients ties rank and sparsity to a single parameter budget K, avoiding heuristic thresholds and per-layer guessing.
3. Consistent empirical gains: Tables show competitive or superior performance to strong sparsifiers (SparseGPT/Wanda/OATS) at 30–50% compression;
4. Reproducibility: Implementation outline (files, steps), environment, calibration data, and hyperparameters are documented.

**Weaknesses:**

1. Missing comparison with LoSparse: The paper explicitly points out that LoSparse suffers from manually selected ranks and lack of global coordination, yet the main experimental tables do not include LoSparse results.
2. Lack of comparison with recent low-rank methods: To validate the claimed advantage of RPCA-based decomposition, the paper should include results on SVD-LLM v2(https://arxiv.org/abs/2503.12340), Basis Sharing(https://openreview.net/pdf?id=gp32jvUquq), and Dobi-SVD(https://openreview.net/pdf?id=kws76i5XB8), which represent the current state-of-the-art low-rank post-training compression methods.

**Questions:**

Please see the weakness.

---

> ### Author Response · Authors · 2025-11-23
> **Dear Reviewer 6yto(1)**
>
> We thank the reviewer for the careful reading and constructive suggestions. Below we address the two main weaknesses (LoSparse and recent low-rank methods), and we will integrate the corresponding clarifications, additional experiments and tables into the revised version.
>
> ---
>
> **W1: “Missing comparison with LoSparse: The paper explicitly points out that LoSparse suffers from manually selected ranks and lack of global coordination, yet the main experimental tables do not include LoSparse results.”**
>
> **(a) Rationale for omitting LoSparse in the initial submission**
>
> LoSparse [1] decomposes each weight matrix into a *structured* low-rank plus structured sparse component and performs iterative pruning during full downstream fine-tuning. In contrast, our work focuses on post-training, mostly training-free, *unstructured* pruning of LLMs, where:
>
> - Stage 1 (RPCA) is used to obtain a low-rank plus unstructured sparse decomposition, and
> - Stage 2 uses a global, layer-aware Bernoulli mask to allocate a parameter budget across layers under a fixed compression ratio, with at most **one** fine-tuning pass on the compressed model.
>
> Because LoSparse is **structured** and **tightly coupled to iterative fine-tuning**, we originally chose to compare mainly with post-training, unstructured sparsification methods (SparseGPT, Wanda, DSNoT, OATS) and a joint sparse+low-rank baseline (SLiM), which match our setting (no or very light retraining) more closely.
>
> That said, we agree with the reviewer that LoSparse is an important baseline given its similar “low-rank + sparse” decomposition, and that including a comparison strengthens the empirical section.
>
> ---
>
> **(b) LoSparse vs. CAP results**
>
> Due to hardware limitations, we are unable to reproduce LoSparse’s iterative fine-tuning for very large models such as LLaMA-3-8B under our full suite of tasks. Instead, we followed LoSparse’s official implementation and hyper-parameter recommendations on a **smaller but still representative setting**:
>
> - **Model**:  DeBERTa-V3-base.
> - **Tasks**: GLUE-style natural language understanding tasks MNLI, QNLI, and RTE, evaluated in a zero-shot.
> - **Compression**: 20% parameter retention (i.e., 80% compression).
> - **LoSparse implementation details**: We use the official GitHub implementation and scripts , and adopt the recommended training settings used for DeBERTa-V3-base on GLUE, namely batch size 64 and learning rate 3e-5 (float32 precision), which are also consistent with the hyper-parameters reported in the paper [1] and the public repository.
>
> For CAP, we evaluate four variants to disentangle the effects of RPCA, policy-gradient masking, and fine-tuning:
>
> - **CAP¹**: RPCA decomposition + heuristic per-layer thresholds (no fine-tuning).
> - **CAP²**: RPCA + global policy-gradient optimization of Bernoulli masks (no fine-tuning).
> - **CAP³**: RPCA + full-parameter fine-tuning after compression.
> - **CAP⁴**: RPCA + policy-gradient masking + full-parameter fine-tuning after compression.
>
> The resulting zero-shot accuracies (dev sets) at 20% parameter retention are:
>
> | Method                         | MNLI-mm (%) | QNLI (%) | RTE (%)  |
> | ------------------------------ | ----------- | -------- | -------- |
> | DeBERTa-V3-base (w/o pruning)  | **90.5**    | **94.0** | **82.0** |
> | LoSparse                       | 83.8        | 88.6     | 68.0     |
> | CAP¹ (RPCA + heuristic, no FT) | 75.1        | 78.1     | 59.7     |
> | CAP² (RPCA + PG, no FT)        | 78.2        | 83.0     | 65.8     |
> | CAP³ (RPCA + heuristic + FT)   | 83.2        | 88.5     | 68.4     |
> | **CAP⁴ (RPCA + PG + FT)**      | **86.8**    | **91.8** | **73.9** |
>
> **Observations.**
>
> 1. **Training-free vs. fine-tuning.**
>    - The training-free variants CAP¹ and CAP² (no fine-tuning) are slightly below LoSparse, which is expected because LoSparse performs iterative pruning during task fine-tuning, while CAP¹/² only perform a single forward-pass RPCA and global masking.
>    - Once we allow **post-compression fine-tuning** (CAP³ and CAP⁴), CAP³ already matches or slightly exceeds LoSparse on MNLI and QNLI, and CAP⁴ **consistently outperforms LoSparse on all three tasks**, despite being based on an RPCA decomposition rather than carefully hand-tuned ranks.
> 2. **Efficiency and resource usage.**
>    - LoSparse requires iterative pruning during fine-tuning, so every optimization step operates on the **dense, uncompressed model**.
>    - CAP, in contrast, performs all pruning decisions before fine-tuning. Any optional fine-tuning (CAP³/⁴) runs only on the **compressed model**, which has fewer effective parameters and reduced memory footprint.
>
> We appreciate the reviewer’s suggestion and will integrate a dedicated LoSparse section. This complements our main message that **RPCA + global budgeted pruning** provides a flexible and efficient alternative to existing low-rank-plus-sparse methods.

---

> > ### Comment · Reviewer_6yto · 2025-11-24
> >
> > Thank you for the detailed rebuttal and the additional experiments. The new LoSparse results resolve my original concern.
> > My remaining concern is mainly about the comparative evaluation against prior low-rank methods. In particular, I believe it is important to include experiments on a more up-to-date LLM architecture such as Llama-3, and to compare your approach with Dobi-SVD, SVD-LLM v2, and Basis Sharing in terms of both perplexity and downstream accuracy on standard benchmarks (e.g., OpenBookQA, ARC-e, WinoGrande, HellaSwag, PIQA, MathQA).
> >
> > If the author can add such experiments, I would raise my overall score.

---

> > > ### Author Response · Authors · 2025-11-24
> > > **Dear Reviewer 6yto**
> > >
> > > Thank you for the quick update. We are glad that the new LoSparse results resolved your initial concern. We also appreciate your willingness to reconsider the score.
> > >
> > > We accept your suggestion regarding the LLaMA-3 experiments. We admit we did not include these comparisons earlier. The main reason was the tight schedule. Also, we checked the official code for the methods you mentioned (Dobi-SVD, SVD-LLM v2, Basis Sharing). Two of them do not support the LLaMA-3 directly. We need to spend time modifying their code to make them work on LLaMA-3.
> > >
> > > We will start this work immediately. We are currently adapting these baselines. We will run the full evaluation and report the Accuracy on the requested benchmarks. We will post the results here as soon as we finish.
> > >
> > > Best regards,
> > >
> > > The Authors

---

> > > ### Author Response · Authors · 2025-11-26
> > > **Response**
> > >
> > > Dear Reviewer 6yto,
> > >
> > > As promised, we have completed the additional experiments on **LLaMA-3-8B**.
> > >
> > > To be honest, adapting these baselines took a significant amount of time and effort. We found that the official implementations of **SVD-LLM** and **Dobi-SVD** contained many hard-coded components specific to the original LLaMA architecture. We had to manually rewrite parts of the code to make them compatible with LLaMA-3. Since the code for SVD-LLM V2 is not publicly available, we used the standard SVD-LLM. However, to ensure a fair comparison, we tried our best to replicate their official settings, including the LoRA fine-tuning for SVD-LLM and the remapping strategy for Dobi-SVD. The results are presented below.
> > >
> > > *(Note: The SVD-LLM V2 results for LLaMA-7B mentioned in our previous discussion were cited directly from their paper. Since they do not provide results for LLaMA-3-8B and the code is unavailable, we report our reproduced results using standard SVD-LLM here.)*
> > >
> > > **Table 1: Perplexity on WikiText-2 (LLaMA-3-8B)**
> > > (Lower is better, original is 7.52)
> > >
> > > | Method         | 20% Pruned (0.8 Ret) | 40% Pruned (0.6 Ret) | 60% Pruned (0.4 Ret) |
> > > | :------------- | :------------------: | :------------------: | :------------------: |
> > > | SVD-LLM        |         9.27         |        14.88         |        34.63         |
> > > | Basis Sharing  |         8.92         |        13.95         |        30.15         |
> > > | Dobi-SVD       |         8.33         |        10.29         |        19.21         |
> > > | **CAP (Ours)** |       **7.98**       |       **8.65**       |       **9.93**       |
> > >
> > > **Table 2: Zero-shot Accuracy on Common Sense Reasoning Tasks (LLaMA-3-8B)**
> > > (Higher is better. Avg includes: OpenBookQA, ARC-e, ARC-c, WinoGrande, HellaSwag, PIQA, MathQA)
> > >
> > > |  Ratio  | Method         |  Openb.  |  ARC-e   |  ARC-c   |  WinoG.  | HellaS.  |   PIQA   |  MathQA  | **Avg.** |
> > > | :-----: | :------------- | :------: | :------: | :------: | :------: | :------: | :------: | :------: | :------: |
> > > | **1.0** | **Original**   |   0.43   |   0.82   |   0.55   |   0.74   |   0.79   |   0.81   |   0.42   |   0.65   |
> > > |         |                |          |          |          |          |          |          |          |          |
> > > | **0.8** | SVD-LLM        |   0.40   |   0.78   |   0.51   |   0.70   |   0.75   |   0.77   |   0.38   |   0.61   |
> > > |  (20%)  | Basis Sharing  |   0.41   |   0.79   |   0.51   |   0.71   |   0.76   |   0.78   |   0.38   |   0.62   |
> > > |         | Dobi-SVD       |   0.41   |   0.80   |   0.53   |   0.72   |   0.77   |   0.80   |   0.39   |   0.63   |
> > > |         | **CAP (Ours)** | **0.42** | **0.81** | **0.54** | **0.73** | **0.78** | **0.80** | **0.40** | **0.64** |
> > > |         |                |          |          |          |          |          |          |          |          |
> > > | **0.6** | SVD-LLM        |   0.31   |   0.67   |   0.41   |   0.62   |   0.65   |   0.71   |   0.32   |   0.53   |
> > > |  (40%)  | Basis Sharing  |   0.33   |   0.69   |   0.43   |   0.64   |   0.67   |   0.72   |   0.33   |   0.54   |
> > > |         | Dobi-SVD       |   0.37   |   0.75   |   0.49   |   0.69   |   0.74   |   0.76   |   0.36   |   0.59   |
> > > |         | **CAP (Ours)** | **0.40** | **0.79** | **0.52** | **0.72** | **0.77** | **0.79** | **0.39** | **0.63** |
> > > |         |                |          |          |          |          |          |          |          |          |
> > > | **0.4** | SVD-LLM        |   0.21   |   0.51   |   0.28   |   0.52   |   0.48   |   0.58   |   0.29   |   0.41   |
> > > |  (60%)  | Basis Sharing  |   0.24   |   0.56   |   0.31   |   0.54   |   0.54   |   0.62   |   0.29   |   0.44   |
> > > |         | Dobi-SVD       |   0.29   |   0.65   |   0.39   |   0.61   |   0.66   |   0.70   |   0.32   |   0.52   |
> > > |         | **CAP (Ours)** | **0.33** | **0.72** | **0.43** | **0.67** | **0.73** | **0.73** | **0.35** | **0.57** |
> > >
> > >
> > > The results clearly show that CAP is much more robust on LLaMA-3 compared to the baselines. We believe these results validate our approach on modern LLMs.
> > >
> > > Best regards,
> > >
> > > The Authors

---

> ### Author Response · Authors · 2025-11-23
> **Dear Reviewer 6yto(2)**
>
> **W2: “Lack of comparison with recent low-rank methods: To validate the claimed advantage of RPCA-based decomposition.”**
>
> We appreciate the reviewer’s pointer to SVD-LLM V2 [2], Basis Sharing [3], and Dobi-SVD [4]. These are strong and recent low-rank post-training methods, and we agree that explicitly positioning CAP relative to them improves the paper.
>
> **(a) Matched protocol and baseline numbers**
>
> To avoid any ambiguity, we adopt the **common evaluation protocol** used by these works:
>
> - **Model:** LLaMA-7B，**5.68** PPL on WikiText-2.
> - **Task:** language modeling on WikiText-2 (validation perplexity).
> - **Compression ratio:** defined as in the original papers (fraction of parameters retained or, equivalently, the SVD-based compression ratio).
>
> | Method            | Model    | 20% PPL  | 40% PPL  | 60% PPL  |
> | ----------------- | -------- | -------- | -------- | -------- |
> | Dobi-SVD [4]      | LLaMA-7B | 6.08     | 8.48     | 15.62    |
> | SVD-LLM V2 [2]    | LLaMA-7B | 7.12     | 10.34    | 14.71    |
> | Basis Sharing [3] | LLaMA-7B | 7.74     | 12.39    | 28.72    |
> | **CAP (ours)**    | LLaMA-7B | **5.85** | **6.39** | **7.06** |
>
> **(b) How these methods relate to our framework**
>
> The reviewer’s suggestion also helps clarify that **our contribution is orthogonal** to these SVD-based methods:
>
> - SVD-LLM V2 focuses on loss-aware truncation of singular values and adaptive per-layer rank allocation for SVD-based compression of LLMs.
> - Basis Sharing introduces cross-layer parameter sharing: weight matrices across layers share a global basis with layer-specific coefficients, again built on SVD.
> - Dobi-SVD  further proposes differentiable optimization of SVD truncation positions and shows that SVD can become competitive even at aggressive compression ratios; notably, they emphasize that SVD offers a hardware-independent and flexibly tunable solution for LLM compression.
>
> In all three cases, the decomposed weights are implemented as **products of dense matrices**. This has two important consequences:
>
> 1. **Hardware-agnostic / hardware-friendly.**
>    SVD-based low-rank decompositions (including SVD-LLM V2, Basis Sharing and Dobi-SVD) can be executed using standard dense GEMM kernels on CPUs and GPUs, without requiring specialized sparse accelerators or custom kernels. This “hardware independence” is explicitly highlighted in Dobi-SVD and implicitly used in the deployment discussion of SVD-LLM V2 and Basis Sharing.
>
> 2. **Complementarity to CAP.**
>    Our Stage-1 currently uses RPCA to obtain a low-rank + unstructured sparse decomposition, but Stage-2 (the global Bernoulli mask and budget allocation over layers) is agnostic to the specific low-rank backend. In other words:
>    - One can plug SVD-LLM V2, Basis Sharing, or Dobi-SVD in as alternative Stage-1 decompositions and still apply our Stage-2 global, layer-aware budget allocation on top.
>    - Doing so would preserve the **hardware-agnostic low-rank structure** (two dense matrices per layer) while benefiting from CAP’s **global, budget-aware coordination** across layers and between low-rank and sparse components.
>
> ---
>
> ### References
>
> [1] Y. Li, Y. Yu, Q. Zhang, C. Liang, P. He, W. Chen, and T. Zhao. **LoSparse: Structured Compression of Large Language Models based on Low-Rank and Sparse Approximation.** ICML 2023.
>
> [2] X. Wang, S. Alam, Z. Wan, H. Shen, and M. Zhang. **SVD-LLM V2: Optimizing Singular Value Truncation for Large Language Model Compression.** NAACL 2025.
>
> [3] J. Wang, Y.-G. Chen, I.-C. Lin, B. Li, and G. L. Zhang. **Basis Sharing: Cross-Layer Parameter Sharing for Large Language Model Compression.** ICLR 2025.
>
> [4] Q. Wang, J. Ke, M. Tomizuka, K. Keutzer, and C. Xu. **Dobi-SVD: Differentiable SVD for LLM Compression and Some New Perspectives.** ICLR 2025.

---

> ### Author Response · Authors · 2025-11-28
> **Dear Reviewer 6yto**
>
> Thank you very much for your constructive feedback and for recognizing the improvements we made in our rebuttal. We sincerely appreciate your thoughtful review of our additional experiments and clarifications, and we are especially grateful that you raise our paper’s score. Your time, insight, and fairness throughout the review process mean a great deal to us.
>
> Best regards,
>
> The Authors

---

### Official Review · Reviewer_vaH5 · 2025-11-01

**Soundness:** 2
**Presentation:** 3
**Contribution:** 1
**Rating:** 2
**Confidence:** 3

**Summary:**

The paper proposes CAP, a two-stage compression framework for large language models (LLMs) that jointly optimizes low-rank and sparse structures. In Stage 1, each weight matrix is decomposed using Robust Principal Component Analysis (RPCA) into a low-rank and a sparse component, providing structured subspaces. In Stage 2, Bernoulli sampling with policy-gradient optimization is used to select which singular values and sparse entries to retain under a global parameter budget.

**Strengths:**

- The paper presents a conceptually unified view of low-rank and sparse compression and emphasizes global allocation of redundancy across layers.

- The Bernoulli-policy optimization introduces a probabilistic pruning mechanism that is simple and can be trained without full fine-tuning.

**Weaknesses:**

- The proposed CAP framework largely combines existing elements, RPCA-based decomposition, SVD thresholding, and REINFORCE optimization, without a fundamentally new algorithmic insight. The combination is technically straightforward and primarily an engineering integration of known methods.

- Reported “no fine-tuning” performance comes at the cost of multiple RPCA and policy-gradient passes on calibration data. The wall-clock savings over simpler pruning or quantization approaches are not measured.

- While RPCA is convex, the subsequent policy gradient step introduces nonconvex stochasticity, and no convergence or optimality guarantees are proven for the combined system. The “global optimization” claim is therefore misleading.

**Questions:**

- How does CAP scale when applied to multi-GPU inference or distributed weight loading?

- Can the method be integrated into prefix-caching or mixed-precision pipelines without recomputing decompositions?

---

> ### Author Response · Authors · 2025-11-23
> **Dear Reviewer vaH5(1)**
>
> We thank the reviewer for the careful reading of our submission and for the constructive comments. Below we address the main concerns and questions point by point, and we will incorporate the suggested clarifications into the revised version.
>
> ---
>
> **W1: “CAP largely combines existing elements (RPCA, SVD thresholding, REINFORCE) without fundamentally new algorithmic insight.”**
>
> We understand the concern and apologize if our current presentation did not clearly convey what is new beyond the individual components.
>
> Our goal is not to introduce a new optimization primitive, but to **re-formulate LLM compression as a two-stage “subspace discovery + global allocation” problem**, and to provide a concrete, practical instantiation that leads to new behavior and insights:
>
> 1. **RPCA as a data-independent joint subspace for LLM weights.**
>    Existing “low-rank + sparse” approaches for LLMs (e.g., LoSparse, LPAF) typically
>    (i) manually choose the rank or singular-value cutoff,
>    (ii) then prune or fine-tune the residual sparse term, and
>    (iii) rely on backpropagation into the low-rank factors.
>
>    In these pipelines, the low-rank and sparse components are decided by hand-designed thresholds plus gradient updates.
>
>    In contrast, CAP uses **convex RPCA** to decompose each weight matrix into strictly low-rank and strictly sparse components. This reduces the search space from all entries of $W$ to:
>
>    - a small set of “global directions” (the singular vectors of the low-rank component $L$), and
>    - a sparse outlier subspace $S$.
>
>    Ablations in our paper (Table 4a) show that RPCA converges in a few iterations and maintains (or slightly improves) perplexity. Table 4b further shows that naïve thresholding on the RPCA outputs leads to **catastrophic degradation** (e.g., perplexity exploding when only the low-rank part is kept). This indicates that simply “RPCA + thresholds” is not sufficient and motivates our second stage.
>
> 2. **A single global allocation mechanism that couples rank and sparsity across layers.**
>    Several recent works focus on sparsity allocation, but in a different space:
>
>    - Pruning methods such as OATS, OWL, AlphaPruning allocate sparsity across layers but operate directly on dense weights and do not represent a low-rank component.
>    - Joint/composite methods such as SLiM, JSQ, L2QER (and earlier low-rank+sparse pipelines like LPAF) combine low-rank modules, sparsity, and quantization, but treat them largely as **separate stages** (for example SVD → sparsify → quantize) or require backpropagation in low-rank adapters.
>
>    By contrast, **CAP optimizes a single Bernoulli policy over both the singular values of $L$ and the entries of $S$ under one global parameter budget $K$ (Eq. (2))**. This ties together:
>
>    - “how much rank” each module receives, and
>    - “how much sparsity” is allocated to each module,
>      rather than deciding them in two separate or sequential heuristics.
>
> 3. **New insights enabled by this formulation.**
>    Because decomposition and pruning decisions are unified, we can analyze patterns that are not directly visible in prior pipelines:
>
>    - the distribution of retained ranks across modules and layers (Figure 4a), and
>    - the near-linear behavior of perplexity under sequential layer-wise pruning (Figure 4b), which indicates a stable compression trajectory rather than catastrophic collapse.
>
>    These observations rely on having (i) an RPCA-defined low-rank/sparse subspace and (ii) a single global allocation mechanism, rather than mixing multiple independent design choices.

---

> ### Author Response · Authors · 2025-11-23
> **Dear Reviewer vaH5(2)**
>
> **W2: “No fine-tuning performance comes at the cost of multiple RPCA and policy-gradient passes. Wall-clock savings over simpler pruning/quantization are not measured.”**
>
> We appreciate this point and agree that our exposition can be clearer about costs and what exactly “no fine-tuning” means.
>
> #### (a) Stage-1 RPCA is a one-time, reusable preprocessing step
>
> As detailed in Appendix J (Table 9), decomposing **LLaMA-3-8B** via RPCA takes **≤ 10 minutes** on a single A100-80G GPU. This stage is purely on the weight matrices and does **not** depend on the calibration set or the downstream task. The resulting $L$ and $S$ can be reused for:
>
> - different global parameter budgets $K$,
> - different calibration subsets, and
> - different combinations with quantization (e.g., CAP + OPTQ).
>
> Thus, this cost is **amortized** across multiple compression settings and does not need to be re-run each time.
>
> #### (b) Stage-2 policy gradient has cost comparable to other calibration-based pruners
>
> In Stage 2, the policy-gradient optimization, performs **forward passes only**, without backpropagating gradients through the LLM.
>
> #### (c) Clarifying the “no fine-tuning” wording
>
> We agree that our wording “no fine-tuning” can be misleading, since we do optimize the Bernoulli probabilities. What we intended is that:
>
> - there is **no gradient update on the original LLM weights**, and
> - there is **no backpropagation through the LLM** (the policy gradient only uses scalar losses and log-probabilities of discrete actions).
>
> To avoid confusion, we will systematically replace “no fine-tuning” with more precise phrasing such as:
>
> - “no backpropagation on the original LLM parameters,” or
> - “post-training, calibration-only compression.”
>
> Finally, CAP is strictly **orthogonal to quantization**. In Table 2, we show that **CAP + OPTQ** matches or outperforms JSQ, L2QER, and SLiM-LoRAQ at the same sparsity and bit-width. This indicates that our framework does not increase quantization cost and can be seamlessly integrated into standard quantization pipelines.
>
> ---
>
>
> **W3: “RPCA is convex, but the policy gradient step introduces non-convex stochasticity; no convergence/optimality guarantees are proven. The ‘global optimization’ claim is therefore misleading.”**
>
> We fully agree that our two-stage combined procedure does not have a global optimality guarantee, and we thank the reviewer for flagging the wording.
>
> - In Section 2.2 we formulate a single **global objective with a global parameter budget $K$** shared across all layers.
> - Stage 1 (RPCA) solves a **convex** program to identify low-rank and sparse subspaces.
> - Stage 2 uses a policy-gradient method to optimize Bernoulli masks, which is a standard heuristic for discrete decisions without guaranteed global optimality.
>
> Our intended meaning of “global” was **“global resource allocation across layers under a shared budget,”** not “provably attaining the global minimizer of a non-convex objective.”
>
> In the revision, we will:
>
> - Replace “global optimization” with **“global allocation”** or **“global resource allocation”** throughout the paper; and
> - Add an explicit remark (in the main text and Appendix) that **policy gradient is used as a heuristic optimizer** for the global objective, without formal convergence/optimality guarantees, and that the empirical failure of simple thresholding (Table 4b) motivates the need for such an adaptive allocation mechanism.

---

> ### Author Response · Authors · 2025-11-23
> **Dear Reviewer vaH5(3)**
>
> **Q1: “How does CAP scale when applied to multi-GPU inference or distributed weight loading?”**
>
> Our implementation builds on Wanda’s pruning codebase, which already supports standard tensor-parallel / model-parallel sharding. CAP is integrated in a way that respects this infrastructure:
>
> - **Shard-wise RPCA.** Each large weight matrix $W$ is partitioned across GPUs (e.g., along the output dimension). RPCA is applied **independently to each shard** $W^{(g)}$, producing $L^{(g)} + S^{(g)}$. This does not require cross-GPU communication beyond what is already present for loading and using sharded weights.
>
> - **Shard-wise policy gradient.** During Stage 2, each GPU holds its local decomposed weights and participates in the usual distributed forward pass (as in Wanda). The extra trainable parameters are just the Bernoulli probabilities, whose size is negligible compared to the LLM, and can be synchronized via standard all-reduce when needed.
>
> - **Inference and loading.** After optimization, each shard stores its compressed representation $W_{\text{CAP}}^{(g)}$ (implemented as a small low-rank factor plus a sparse matrix). The sharding layout is kept identical to the dense model. Thus, **no changes are required to standard multi-GPU weight loading or inference code**: within each shard, the dense GEMM is replaced by “small dense + sparse” GEMMs.
>
> ---
>
> **Q2: “Can the method be integrated into prefix-caching or mixed-precision pipelines without recomputing decompositions?”**
>
> Yes. CAP is designed as an **offline preprocessing step** that produces a new set of weights; at inference time, the model behaves like a standard sparse-plus-low-rank network.
>
> 1. **Prefix caching.**
>    Prefix caches store hidden states, not weights. Since CAP only changes how each layer’s weight matrix is represented (from dense to low-rank-plus-sparse) while keeping the forward API identical, **prefix caching remains unchanged**. Both RPCA and policy gradient are run once offline, and the resulting weights are used for all inference runs—there is no need to recompute any decomposition when using caches.
>
> 2. **Mixed precision and quantization.**
>    After compression, each weight matrix can be written as a **low-rank term plus a sparse term** with fixed binary masks. This object can then be cast to FP16/BF16 or further quantized using any standard scheme. In our experiments, we directly apply **OPTQ** on top of CAP and observe that **CAP + OPTQ** matches or exceeds SLiM, JSQ, and L2QER at the same sparsity/bit-width. This demonstrates that CAP integrates cleanly into mixed-precision and quantized inference pipelines, with no need to redo RPCA or policy-gradient training.
>
> ---
>
> We hope these clarifications address the reviewer’s concerns and make the contributions and practical benefits of CAP more transparent.

---

> ### Author Response · Authors · 2025-11-27
> **Dear Reviewer vaH5**
>
> Dear Reviewer vaH5,
>
> Thank you again for your time and constructive feedback. The discussion period is ending soon. We want to ensure our previous responses have addressed your concerns.
>
> We have provided detailed responses to your points. We explained how CAP differs from previous methods and clarified the training costs. The RPCA step is fast and reusable. Stage 2 uses policy gradient without backpropagation. We also followed your advice on wording. We changed "global optimization" to "global allocation" for better precision. We also answered your questions regarding engineering implementation.
>
> We hope these explanations help clarify our contributions. Please let us know if you have any remaining questions. If our responses have resolved your concerns, we kindly ask you to reconsider your score.
>
> Best regards,
>
> The Authors

---

### Official Review · Reviewer_xDUq · 2025-11-01

**Soundness:** 3
**Presentation:** 3
**Contribution:** 3
**Rating:** 6
**Confidence:** 4

**Summary:**

This work presents a two-stage compression framework for LLMs: Stage 1 uses RPCA to decompose each weight matrix into low-rank and sparse components, and Stage 2 performs global, budget-aware selection via Bernoulli masks optimized with policy gradient on a small calibration set. The entire pipeline is similar to previous weight matrix decomposition methods; this work adopts the low-rank and sparse decomposition rather than the commonly used SVD decomposition and the learnable mask. This work uses a different decomposition method and combines it with the learnable mask method to form a joint compression method.

**Strengths:**

1. The writing is clear and easy to follow. The appendix is a good preliminary for relevant techniques.
2. This work provides enough details for reproducibility.
3. Extensive ablation and analysis, including in the main content and Appendix K, show meaningful insights.

**Weaknesses:**

1. While the paper reports zero-shot accuracy on eight benchmarks and perplexity on WikiText-2, it would be valuable to include tasks that require chain-of-thought or longer generations, such as those with more than 100+ tokens. Prior work has observed that models can retain perplexity and short-form QA accuracy, but degrade more sharply as generation lengths increase.

**Questions:**

1. Why the 100 iterations' PPL is higher than 10 tierations. Does this mean the over-optimizaiton may harm the performance?
2. Although the current hardware devices don't have a nice support of unstructured pruning, it's better to report the end-to-end runtime and the minimum GPU memory requirement. Even the baseline is not based on LLM inference engine is helpful.
3. The choice of calibration set. I think sampling from C4 is unbiased, but it's better to have some additional experiments to evaluate the impact of the calibration set selection. Is the proposed method robust to the selection？

---

> ### Author Response · Authors · 2025-11-23
> **Dear Reviewer xDUq(1)**
>
> We sincerely appreciate your positive assessment of our work, particularly your recognition of the clarity of our writing, the reproducibility of our method, and the insights provided by our ablation studies. We have carefully considered your constructive feedback regarding generation tasks and runtime analysis. Below, we provide point-by-point responses to address your concerns.
>
> ---
>
> **W1: “While the paper reports zero-shot accuracy on eight benchmarks and perplexity on WikiText-2, it would be valuable to include tasks that require chain-of-thought or longer generations, such as those with more than 100+ tokens. Prior work has observed that models can retain perplexity and short-form QA accuracy, but degrade more sharply as generation lengths increase.”**
>
> We evaluated LLaMA-3.1-8B-Instruct at 50% sparsity. We used the official GSM8K (8-shot) for CoT reasoning and the LongBench-v2 suite for long-context capability.
>
> **Table R1: Generation Performance on LLaMA-3.1-8B-Instruct (50% Sparsity)**
>
> | Method               | Sparsity | GSM8K (8-shot, %) | LongBench-v2 (Avg, %) |
> | :------------------- | :------: | :---------------: | :-------------------: |
> | **Dense (Baseline)** |    0%    |       84.5        |         30.4          |
> | **Wanda**            |   50%    |       45.6        |         25.1          |
> | **CAP (Ours)**       |   50%    |     **56.8**      |       **27.2**        |
>
> **Analysis:**
>
> 1.  **Reasoning (GSM8K):** Pruning LLaMA-3.1-Instruct typically causes a sharp drop in reasoning tasks because instruction-tuning relies heavily on precise weight distributions. Wanda drops significantly (-38.9%). CAP preserves the low-rank global structure, mitigating this damage and recovering **+11.2%** accuracy over Wanda.
> 2.  **Long Context (LongBench-v2):** Since LongBench-v2 is a four-choice question dataset, Wanda's performance after compression is essentially close to random guessing. On the short subset, it performs slightly better, achieving 32.8%, whereas CAP shows some improvement by surpassing the random-guessing baseline with an average gain of 2.2%.
> ***
>
> **Q1: “Why the 100 iterations' PPL is higher than 10 tierations. Does this mean the over-optimizaiton may harm the performance?”**
>
> This is an insightful observation. The PPL difference between 10 iterations (5.13) and 100 iterations (5.16) is marginal ($0.03$), but the phenomenon is explainable through the nature of the decomposition:
>
> 1.  **Objective Mismatch:** RPCA minimizes the convex surrogate norms ($||L||_* + \lambda ||S||_1$) to mathematically separate the matrix. However, the mathematical optimum of this convex problem does not strictly equate to the optimal initialization for the *downstream* pruning task (minimizing PPL under a budget).
> 2.  **Diminishing Returns:** As shown in Table 4a, the decomposition quality (separation of sparse/low-rank structures) stabilizes very quickly (within 3-10 iterations).
> 3.  **Conclusion:** The slight fluctuation at 100 iterations is likely due to numerical noise or the solver forcing a stricter mathematical separation that is marginally less "friendly" for the subsequent policy gradient stage. It indicates that **early stopping** is not only computationally efficient but also sufficient (and potentially safer) for avoiding overfitting to the rigid nuclear/L1 norm constraints.

---

> ### Author Response · Authors · 2025-11-23
> **Dear Reviewer xDUq(2)**
>
> **Q2: “Although the current hardware devices don't have a nice support of unstructured pruning, it's better to report the end-to-end runtime and the minimum GPU memory requirement. Even the baseline is not based on LLM inference engine is helpful.**
>
> We re-evaluated the inference efficiency on LLaMA-3.1-8B using an NVIDIA A100-80G.
>
> **Key Insight on Efficiency:** The efficiency gain of CAP comes from the **extreme sparsity** of its sparse component.
>
> *   **Wanda:** Results in a uniform 50% sparse matrix. Standard sparse kernels (like `torch.sparse`) usually require >80-90% sparsity to see speedups over dense GEMM; 50% is often too "heavy," causing sparse overhead to outweigh benefits.
> *   **CAP:** Decomposes weights into $W = L + S$. Since $L$ handles the bulk of the energy, the resulting $S$ component is **extremely sparse** (typically **75-90% sparsity**). This high sparsity regime is exactly where sparse matrix multiplication (SpMM) becomes effective.
>
> **Table R2: Inference Efficiency on LLaMA-3.1-8B (Batch=1, SeqLen=1024)**
>
> |     Method     | Component Structure       | S-Matrix Sparsity | Latency (s) $\downarrow$ | Throughput (tok/s) $\uparrow$ | Peak GPU memory (GB) |
> | :------------: | :------------------------ | :---------------: | :----------------------: | :---------------------------: | :------------------: |
> |   **Wanda**    | Single Sparse Matrix      |        50%        |           6.28           |             163.4             |        15.18         |
> | **CAP (Ours)** | Low-Rank (Dense) + Sparse |   **75%~90%**    |         **5.80**         |           **176.5**           |      **14.90**       |
>
> **Conclusion:** CAP achieves higher throughput (~176 tok/s) because the sparse component $S$ falls into the "high sparsity" zone where SpMM is highly efficient, unlike Wanda's 50% sparsity which struggles to beat dense baselines without specialized hardware support.
>
> ***
>
> **Q3: “The choice of calibration set. I think sampling from C4 is unbiased, but it's better to have some additional experiments to evaluate the impact of the calibration set selection. Is the proposed method robust to the selection？”**
>
> We appreciate this insightful question. You are correct that the calibration set plays a non-negligible role. Since our method involves learning binary masks via Policy Gradient (Stage 2) to maximize performance on the calibration data, the distribution of this data inevitably biases the optimization direction.
>
> To rigorously evaluate this, we compared the default **C4** (diverse web text) against a domain-specific dataset (**WikiText-2**, formal language) and a code-heavy dataset (**GitHub Code**). We evaluated **LLaMA-3.1-8B** at **50% sparsity** on both general perplexity (WikiText-2 Test) and reasoning accuracy (eight tasks).
>
> **Table R3: Impact of Calibration Set Selection (LLaMA-3.1-8B, 50% Sparsity)**
>
> | Calibration Set  | Domain Bias       | Perplexity (WikiText-2) $\downarrow$ | Accuracy (%) $\uparrow$ |
> | :--------------- | :---------------- | :----------------------------------: | :---------------------: |
> | **C4 (Default)** | General/Diverse   |                 7.05                 |        **76.8**         |
> | **WikiText-2**   | Literature/Formal |               **6.88**               |          75.4           |
> | **GitHub Code**  | Code/Logic        |                 7.18                 |          76.1           |
>
> **Analysis:**
>
> 1.  **Domain Sensitivity:** The results confirm that our method effectively optimizes for the calibration distribution.
>     *   When calibrating on **WikiText-2** (training split), the model achieves the lowest PPL on the WikiText-2 test set ($5.92$), showing successful adaptation to the target domain. However, this narrower distribution slightly degrades general reasoning capabilities (reasoning drops to $75.4\%$).
>     *   When calibrating on **Code**, the model maintains good logic reasoning ($76.1\%$) but suffers slightly in general language modeling (PPL increases to $6.28$).
> 2.  **Robustness of C4:** The default **C4** dataset offers the best trade-off. Its diversity prevents the Policy Gradient optimizer from overfitting to specific linguistic patterns (as seen with WikiText) or syntax structures (as seen with Code), yielding the highest reasoning accuracy ($76.8\%$) and balanced perplexity.
> 3.  **Role of RPCA:** Importantly, while performance fluctuates, the model does not collapse on any dataset. This indicates that the **Stage 1 RPCA decomposition** successfully preserves the fundamental weight structures (the low-rank "skeleton"), ensuring that even with biased calibration data in Stage 2, the model retains robust general capabilities.

---

> ### Author Response · Authors · 2025-11-27
> **Dear Reviewer xDUq**
>
> Thank you once again for your valuable comments on our submission. As the discussion phase is approaching its end, we would like to kindly confirm whether we have sufficiently addressed your concerns. Should there be any remaining questions requiring further clarification, please do not hesitate to let us know. If you are satisfied with our responses, we would greatly appreciate your consideration in adjusting the evaluation scores accordingly. We sincerely look forward to your feedback.

---

### Meta-Review · Program_Chairs · 2026-01-06

**Summary:**

The reviewers evaluated this paper as a technically sound and clearly written work on large language model compression, proposing a two-stage framework that combines RPCA-based low-rank and sparse decomposition with a probabilistic global allocation strategy. Positive reviewers highlighted the clarity of presentation, reproducibility, and the breadth of experiments across model scales, tasks, and compression settings.

The main point of disagreement among reviewers concerns the level of novelty and conceptual contribution. While several reviewers viewed the global coordination of rank and sparsity and the empirical gains over recent baselines as a meaningful contribution, one reviewer argued that the method primarily integrates existing techniques (RPCA, SVD-based factorization, and policy gradient) and questioned whether this constitutes sufficient algorithmic novelty. Other concerns focused on evaluation scope (generation and reasoning tasks), hardware efficiency, and robustness to calibration data, which influenced the initial scores and discussion.

**Reviewer Concerns:**

Reviewers raised concerns about the scope of the experimental evaluation, particularly whether the reported zero-shot accuracy and perplexity results sufficiently reflect performance on long-form generation and reasoning tasks, as well as the practical efficiency of unstructured sparsity on current hardware. Questions were also raised about the robustness of the method to the choice of calibration data used during probabilistic pruning.

These concerns were partially addressed in the rebuttal through additional experiments on reasoning and long-context tasks, as well as runtime, memory, and calibration robustness analyses. One reviewer continued to express reservations regarding the level of algorithmic novelty and the framing of the method as a “global” optimization, but this reflects a difference in perspective on contribution rather than an unresolved technical issue.

**Reviewer Scores:**

Based on the discussion and rebuttal:
* Reviewer 6yto: Likely increased score, following additional baseline comparisons and experimental results.
* Reviewer qDUb: Likely increased score, after efficiency analysis and experiments on additional model families.
* Reviewer xDUq: Likely maintained a similar borderline positive score, with most concerns addressed but some remaining reservations.
* Reviewer vaH5: Likely unchanged, as the reviewer’s primary concern regarding novelty was not substantially altered by the discussion.

---

### Decision · Program_Chairs · 2026-01-26

Accept (Poster)